# The Price of Freedom: Exploring Expressivity and Runtime Tradeoffs in Equivariant Tensor Products

**YuQing Xie** [1]  **Ameya Daigavane** [1]  **Mit Kotak** [2]  **Tess Smidt** [1]

## Abstract

$E(3)$-equivariant neural networks have demonstrated success across a wide range of 3D modelling tasks. A fundamental operation in these networks is the tensor product, which interacts two geometric features in an equivariant manner to create new features. Due to the high computational complexity of the tensor product, significant effort has been invested to optimize the runtime of this operation. For example, Luo et al. (2024) recently proposed the Gaunt tensor product (GTP) which promises a significant speedup. In this work, we provide a careful, systematic analysis of a number of tensor product operations. In particular, we emphasize that different tensor products are not performing the same operation. The reported speedups typically come at the cost of expressivity. We introduce measures of expressivity and interactability to characterize these differences. In addition, we realized the original implementation of GTP can be greatly simplified by directly using a spherical grid at no cost in asymptotic runtime. This spherical grid approach is faster on our benchmarks and in actual training of the MACE interatomic potential by 30%. Finally, we provide the first systematic microbenchmarks of the various tensor product operations. We find that the theoretical runtime guarantees can differ wildly from empirical performance, demonstrating the need for careful application-specific benchmarking. Our code is available at https://github.com/atomicarchitects/PriceofFreedom.

*Equal contribution [1]Department of EECS, Massachusetts Institute of Technology, Cambridge MA 02139, USA [2]Department of CSE, Massachusetts Institute of Technology, Cambridge MA 02139, USA. Correspondence to: YuQing Xie <xyuqing@mit.edu>, Tess Smidt <tsmidt@mit.edu>.

*Proceedings of the $42^{nd}$ International Conference on Machine Learning*, Vancouver, Canada. PMLR 267, 2025. Copyright 2025 by the author(s).

## 1. Introduction

Many complex physical systems possess inherent spatial symmetries, and incorporating these symmetries into models has been shown to significantly improve both learning efficiency and robustness (Batzner et al., 2022; Rackers et al., 2023; Frey et al., 2023; Owen et al., 2024). To address the specific symmetries present in 3D systems, considerable effort has been dedicated to the development of $E(3)$-equivariant neural networks (E(3)NNs) (Thomas et al., 2018; Weiler et al., 2018; Kondor, 2018; Kondor et al., 2018). E(3)NNs have delivered strong performance across a wide range of scientific applications, including molecular force fields (Batzner et al., 2022; Musaelian et al., 2023; Batatia et al., 2022), catalyst discovery (Liao & Smidt, 2023), generative models (Hoogeboom et al., 2022), charge density prediction (Fu et al., 2024), and protein structure prediction (Lee et al., 2022; Jumper et al., 2021).

The group $E(3)$ consists of all rotations, translations and reflections in 3 dimensions. A function $f : X \to Y$ is $E(3)$-equivariant if it satisfies:

$$f(g \cdot x) = g \cdot f(x) \quad \forall\, g \in E(3), x \in X \qquad (1)$$

where the group action $\cdot$ may differ on the input space $X$ and output space $Y$.

$E(3)$-equivariant neural networks work with features that transform as irreducible representations of $O(3)$, termed 'irreps', as described in Appendix B. How these irreps transform under 3D rotations (elements of $SO(3)$) is defined by a positive integer $L$, which can intuitively be thought of as an angular frequency. As described in Section 2, tensor products are used to construct equivariant bilinearities, a key component of $E(3)$ networks.

The only true tensor product is the Clebsch-Gordan tensor product (CGTP) which uses the well-studied Clebsch-Gordan (Varshalovich et al., 1988) coefficients. However, it has a time complexity[1] of $\mathcal{O}(L^5)$ as we show in Appendix E, which can quickly become expensive for larger $L$. This scaling limits the direct application of $E(3)$-equivariant neural

---

[1]Note that Passaro & Zitnick (2023) claims a runtime of $\mathcal{O}(L^6)$ for this tensor product. In Appendix E, we show that this runtime can be reduced to $\mathcal{O}(L^5)$.

networks to larger systems. Indeed, a number of works suggest that incorporating higher $L$ features can be crucial to improving model performance (Cen et al.; Frank et al., 2022; Aykent & Xia). Hence, there is significant interest in optimizing the tensor product.

One optimization was identified by Passaro & Zitnick (2023) in the special case where one input is a projection of a single vector onto spherical harmonics, an operation commonly used in 3D graph convolutions. Under suitable rotation, these irreps become sparse, allowing for a runtime of $\mathcal{O}(L^3)$. However, the extreme sparsity is not generally true for arbitrary irrep values.

For arbitrary irrep values, Luo et al. (2024) proposed the Gaunt Tensor Product (GTP) which they show has a complexity $\mathcal{O}(L^3)$. Further, Unke & Maennel (2024) introduced another $\mathcal{O}(L^3)$ operation which we call matrix tensor product (MTP). While this represents exciting progress, it raises an important question: *What is fundamentally different between these new tensor products and the general CGTP with $\mathcal{O}(L^5)$ complexity?*

In this paper, we provide a framework for systematically analyzing these new operations. Importantly, we clarify that most new proposed 'tensor products' are *not tensor products* in the mathematical sense, and propose to call them *tensor product operations* (TPOs) instead. We organize this paper by first motivating TPOs in Section 2. We then define a measure of expressivity and interactability for TPOs. In Section 3, we introduce a number of TPOs, and we summarize the asymptotic runtimes and expressivities of various TPO implementations in Section 4. Finally, in Section 5 we benchmark the various tensor products showing that asymptotics do not always correspond with practical performance.

We summarize our core contributions as follows:

- A measure for expressivity and interactability of TPOs.

- A simpler implementation of GTP using projection onto the sphere $S^2$, which has the same asymptotics but is faster in practice.

- A comprehensive analysis of asymptotic runtimes and expressivity of different classes of TPOs.

- Rigorous benchmarks of various TPO implementations.

We assume familiarity with group representations of $SO(3)$. For a brief introduction to representations (reps) and irreducible representations (irreps) of $SO(3)$, we refer the reader to Appendix B.

## 2. Analyzing Tensor Product Operations

### 2.1. Motivating Irreps

We begin by motivating why many equivariant architectures (Geiger & Smidt, 2022; Unke & Maennel, 2024) use irreps to represent features. In particular, it is much easier to enforce equivariance when our features are in terms of irreps because they facilitate parameterization of the most general equivariant linear maps.

First, while there are an infinite number of ways to choose finite dimensional representations, we can always decompose them into a direct sum of irreps. Irreps are hence the fundamental building blocks of arbitrary representations and they are well studied for many groups. Further, assuming we are working over an algebraically closed field, linear layers between a pair of irreps must either be 0 if they are inequivalent or multiples of the identity if they are equivalent. This is known as Schur's lemma (Dresselhaus et al., 2007). Hence once we rewrite representations explicitly as a direct sum of irreps, parameterizing equivariant linear layers becomes trivial. This is why identifying features in terms of irreps is so prevalent in equivariant architectures.

However, Schur's lemma explicitly prevents linear mixing of inequivalent irreps. Hence we need appropriate nonlinear interactions. The simplest possibility is to use multiplication, motivating the tensor product.

### 2.2. Tensor Products and Bilinearities

Here we provide a unified framing for understanding why tensor products are expensive and how other works attempt to remedy this problem. In particular we want to emphasize the connection to constructing bilinearities and that other 'tensor products' are fundamentally different operations from the formal mathematical tensor product.

Given two vector spaces $X, Y$, one can construct a tensor product space $X \otimes Y$ with an associated bilinear mapping $T : X \times Y \to X \otimes Y$. This bilinear mapping is often referred to as the tensor product. In practice, if we have a basis $x_1, \ldots, x_m$ and $y_1, \ldots, y_n$ of $X, Y$ respectively, then we usually use a basis $T(x_i, y_j)$ (often written as $x_i \otimes y_j$) for the tensor product space $X \otimes Y$.

Now suppose we have a space $Z$. Then in practice, we typically parameterize a linear layer $\mathsf{Lin} : X \otimes Y \to Z$. The composition $\mathsf{Lin} \circ T : X \times Y \to Z$ is a bilinearity $X \times Y \to Z$. Note that this construction is *universal*, as all possible bilinearities can be written as $\mathsf{Lin} \circ T$ for some choice of linearity $\mathsf{Lin}$.

While tensor products allow interactions between different irreps and simple construction of all possible bilinearities, they are expensive. First, the dimension of $X \times Y$ is $\mathcal{O}(|X||Y|)$ where $|X|$ and $|Y|$ are the dimensions of $X$ and

$Y$ respectively. This is in fact a lower bound on asymptotic runtime. Further, even if $X, Y$ are explicitly written in a basis which is a direct sum of irreps, the most natural basis of $X \otimes Y$ is generally not. Hence we must perform a change of basis of $X \otimes Y$ into one which explicitly is a direct of irreps. The change of basis matrix is known as the Clebsch-Gordan coefficients. Naively, it requires $\mathcal{O}(|X|^2|Y|^2)$ time to perform this change of basis. The tensor product whose output is explicitly written in an irrep basis is the Clebsch-Gordan tensor product.

To optimize the tensor product, one can leverage the structure of the Clebsch-Gordan coefficients (such as sparsity) to reduce the $\mathcal{O}(|X|^2|Y|^2)$ cost. Still, such approaches can never beat the lower bound of $\mathcal{O}(|X||Y|)$. In typical analysis,[2] $|X|, |Y|$ are $\mathcal{O}(L^2)$ making the lower bound as $\mathcal{O}(L^4)$. However, other 'tensor products' claim asymptotic times lower than $\mathcal{O}(L^4)$ while still demonstrating good performance (Luo et al., 2024; Unke & Maennel, 2024). Fundamentally, what these operations do is replace $T : X \times Y \to X \otimes Y$ with some other bilinearity $T : X' \times Y' \to Z'$. While these bilinearities are often called 'tensor products', they are *not truly tensor products in the mathematical sense*. In this paper, we will refer to these bilinearities as tensor product operations (TPOs).

**Definition 2.1** (Tensor product operations). Let $X', Y', Z'$ be vector spaces equipped with actions of $G$. We refer to any equivariant bilinear map $T : X' \times Y' \to Z'$ as a tensor product operation.

### 2.3. Expressivity and Construction of Bilinearities

It turns out that the runtime savings of alternative TPOs can be understood as coming from a reduction in expressivity. Motivated by the use of tensor product operations to construct bilinearities, we use the *dimension of constructible bilinearities* as a proxy for expressivity.

To do so, we must first understand how to use TPOs to construct a bilinearity. Similar to tensor products, we use equivariant linear layers; however, now we can also add linear layers for the inputs[3]. Hence, we define:

$$\mathsf{Lin}_{\theta_X} : X \to X' \quad \mathsf{Lin}_{\theta_Y} : Y \to Y' \quad \mathsf{Lin}_{\theta_Z} : Z' \to Z$$

parameterized by $\theta = (\theta_X, \theta_Y, \theta_Z) \in \Theta$. Composing these operations, we obtain a bilinearity $B_{T,X,Y,Z,\theta} : X \times Y \to Z$ defined as:

$$B_{T,X,Y,Z,\theta}(\mathbf{x}, \mathbf{y}) = \mathsf{Lin}_{\theta_Z} T(\mathsf{Lin}_{\theta_X}\mathbf{x}, \mathsf{Lin}_{\theta_Y}\mathbf{y}). \quad (2)$$

---

[2]Usually, $X$ transforms as the direct sum of all irreps up to some $L$. As each irrep has dimension $2L+1$, the dimension of $X$ is $|X| = \sum_{l=0}^{L} 2l + 1 = (L+1)^2 = \mathcal{O}(L^2)$.

[3]In the case of the CGTP, adding equivariant linear layers for the inputs is redundant, as the weights can be absorbed into $\mathsf{Lin}_{\theta_Z}$. Hence, we only use an equivariant linear layer $\mathsf{Lin}_{\theta_Z}$ for the outputs in Section 2.2.

We can now naturally define the expressivity of a given tensor product by how many equivariant bilinearities it can parametrize.

**Definition 2.2** (Expressivity of TPOs). Given a TPO $T : X' \times Y' \to Z'$, the space $\mathcal{B}_{T,X,Y,Z} = \{B_{T,X,Y,Z,\theta} : \forall \theta \in \Theta\}$ is the set of all bilinear maps we can construct by inserting equivariant linear layers in the inputs ($\mathsf{Lin}_{\theta_X}$, $\mathsf{Lin}_{\theta_Y}$) and outputs ($\mathsf{Lin}_{\theta_Z}$) in Equation 2. We define the **expressivity** of $T$ with respect to $X, Y, Z$ as the *dimension* of $\mathcal{B}_{T,X,Y,Z}$.

In Section 4, we apply this expressivity measure to various tensor product operations.

### 2.4. Interactability and Selection Rules

A natural question to ask about the constructed bilinearities is which inputs can affect which outputs. Since equivariant linear layers can mix irreps of the same type together, answering this question for bilinearities between single irreps suffices. Using our definition of expressivity, we provide a natural definition for interactability.

**Definition 2.3** (Interactability). Let $T : X' \times Y' \to Z'$ be a TPO. Let $(\ell_1, \ell_2, \ell_3)$ be a triple of irrep types. Define vector spaces $V^{\ell_1}, V^{\ell_2}$ and $V^{\ell_3}$ which transform under irrep types $\ell_1, \ell_2$ and $\ell_3$ respectively. We say $[\ell_1, \ell_2, \ell_3]$ is **interactable** if the expressivity of $T$ with respect to $V^{\ell_1}, V^{\ell_2}, V^{\ell_3}$ is nonzero.

We can analyze interactability by directly analyzing the TPO $T$. Suppose the irreps within $X', Y', Z'$ are labeled by the tuples $(\ell_X, c_X), (\ell_Y, c_Y)$ and $(\ell_Z, c_Z)$, where $\ell$ indicates the irrep type and $c$ is an index over its multiplicity. A **selection rule** for $T$ is a condition on the labels $(\ell_X, c_X), (\ell_Y, c_Y), (\ell_Z, c_Z)$ which must be satisfied for

$$T|_{(\ell_X,c_X),(\ell_Y,c_Y),(\ell_Z,c_Z)} : X'^{(\ell_X,c_X)} \times Y'^{(\ell_Y,c_Y)} \to Z'^{(\ell_Z,c_Z)}$$

to be nonzero. Importantly, note that satisfying a selection rule is a necessary but not sufficient condition for the restricted bilinearity to be nonzero.

The reason selection rules are useful is that if there is no choice of $(c_X, c_Y, c_Z)$ such that $(\ell_X, c_X), (\ell_Y, c_Y)$ and $(\ell_Z, c_Z)$ satisfy the selection rules for a given $T$, then $(\ell_1, \ell_2, \ell_3)$ is not interactable under $T$. Hence, analyzing selection rules helps formally characterize which interactions are excluded by a specific TPO.

### 2.5. Generalizations to $\nu$-fold tensor products and non-irrep basis

We would like to also mention that there have been some recent works dealing with a Cartesian tensor basis (Shao et al., 2024; Zaverkin et al.). Specifying a fully expressive equivariant linear layer is more difficult for such representations, however, the corresponding tensor product is usually simpler

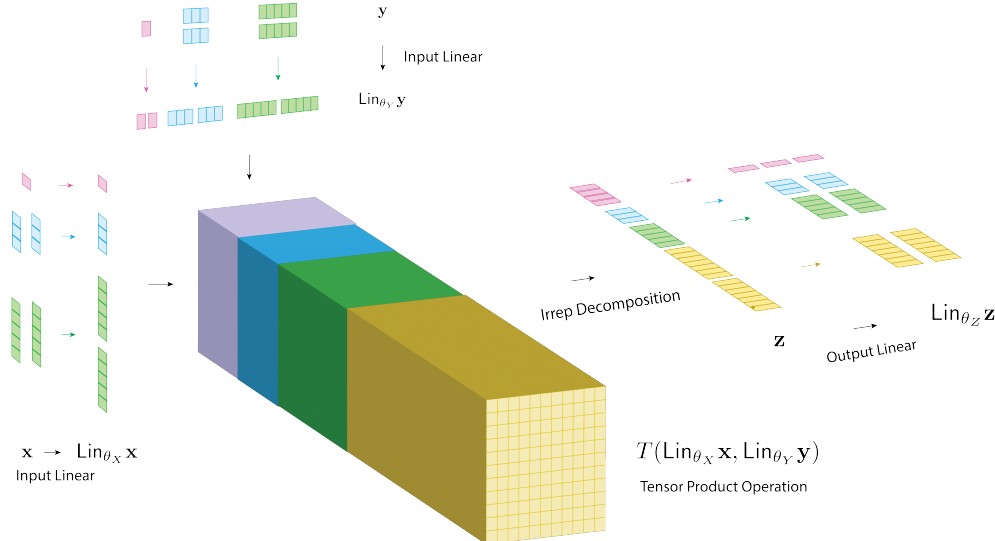

*Figure 1.* Overall schematic of an equivariant bilinearity, where the two inputs **x** and **y** are passed through linear layers and combined using the tensor product operation $T$ to form **z**. The output irreps **z** are passed through a final linear layer. In the case of the CGTP, this would consist of elementwise multiplication of **x** and **y** followed by contraction with the Clebsch-Gordan coefficients to form output irreps **z** . Each irrep of a particular type is denoted by its own color. Linear layers $\mathrm{Lin}_{\theta_X}$, $\mathrm{Lin}_{\theta_Y}$, $\mathrm{Lin}_{\theta_Z}$ can only map between irreps of the same type (indicated by arrows of the corresponding color). However, $T$ can create irreps of a different type (shown in yellow).

to compute. In particular, computing multiple $\nu$-fold tensor products in succession seems to be asymptotically more efficient in this basis (Zaverkin et al.). However, most architectures only use 2-fold tensor products for which Cartesian based approaches have poor asymptotic scaling (Zaverkin et al.). Hence, our analysis focuses on the 2-fold case with an irrep basis.

In principle, our definitions of expressivity and interactability can generalize for $\nu$-fold tensor products as well. We simply replace the TPO which is a fixed bilinearity with a fixed $\nu$-linearity in Definitions 2.2 and 2.3 and similarly consider attaching equivariant linear layers to the inputs and outputs of the $\nu$-linearity.

For alternative choices of basis, we may incur additional costs in the construction of equivariant linear layers. In the case of irrep basis, this construction is cheap aaand so we focus on the asymptotic runtime of the TPO. However for other basis choices, this may not be true and one must take care in analyzing the cost of constructing equivariant linear layers for a fair comparison.

# 3. Tensor Product Operations

## 3.1. Clebsch-Gordan Tensor Product

The Clebsch-Gordan Tensor Product (CGTP) is the only TPO considered which is actually a tensor product. The change of basis of tensor product reps into irreps is well

studied. The corresponding selection rules are well known (Varshalovich et al., 1988). For simplicity, we describe them assuming a single copy of each irrep in the inputs.

**Proposition 3.1** (Selection rule for CGTP)**.** Suppose we have irrep labels $(\ell_1, 1), (\ell_2, 1)$, and $(\ell_3, c_Z)$. We must have $\ell_a \leq \ell_b + \ell_c$ for all choices of distinct $a, b, c \in \{1, 2, 3\}$ and $c_Z = (\ell_1, \ell_2)$.

Importantly, interactions which can form a $\ell_3$ irrep are placed in separate channels, which means that there can be multiple $\ell_3$ irreps created as a result of the CGTP. We refer to any triple $[\ell_1, \ell_2, \ell_3]$ for which $(\ell_1, 1), (\ell_2, 1), (\ell_3, (\ell_1, \ell_2))$ satisfies the selection rule for CGTP as a valid **path**.

## 3.2. Gaunt Tensor Product

The Gaunt tensor product operation (GTP) as introduced by Luo et al. (2024) is a bilinearity

$$(0 \oplus \ldots \oplus L) \times (0 \oplus \ldots \oplus L) \to (0 \oplus \ldots \oplus 2L).$$

It uses the intimate connection between spherical harmonics, $SO(3)$ irreps, and spherical signals, as defined more precisely in Appendix C. The idea is that any rep of form $(0, \ldots, L)$ can be interpreted as coefficients for the spherical harmonics, and hence, corresponds to a function on $S^2$.

In particular, given two $(0, 1, \ldots, L)$ reps **x** and **y**, let $f_{\mathbf{x}} = \mathrm{ToSphere}(\mathbf{x})$ and $f_{\mathbf{y}} = \mathrm{ToSphere}(\mathbf{y})$ be the associated signals on $S^2$. Taking the pointwise product of $f_{\mathbf{x}}$ and $f_{\mathbf{y}}$

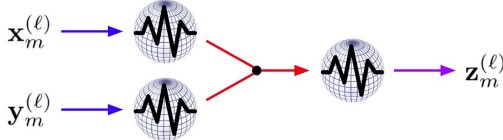

*Figure 2.* Schematic of GTP. We interpret input irreps as scalar SH coefficients to create spherical signals. We then take pointwise products of the two signals to create a new signal which we decompose back into scalar SH coefficients.

on $S^2$ gives us a new function $f_\mathbf{x} \cdot f_\mathbf{y}$, also on $S^2$. Then, converting back to irreps gives us the Gaunt tensor product:

$$\mathbf{x} \otimes_{\text{GTP}} \mathbf{y} = \text{FromSphere}(f_\mathbf{x} \cdot f_\mathbf{y}) \qquad (3)$$

Selection rules for GTP can be derived from the Gaunt coefficients (Gaunt, 1929). Note that there is necessarily only a single copy of each irrep type, even in the output.

**Proposition 3.2** (Selection rules for GTP). For irrep labels $(\ell_1, 1), (\ell_2, 1), (\ell_3, 1)$, the corresponding interaction is nonzero only if the following are satisfied:

- $\ell_a \leq \ell_b + \ell_c$ for any distinct $a, b, c \in \{1, 2, 3\}$.

- $\ell_1 + \ell_2 + \ell_3$ is even.

Note in particular a cross product corresponds to the $[1, 1, 1]$ path which fails rule 2. This formalizes the intuition that GTP is a *symmetric operation* and is unable to perform antisymmetric interactions. In Section 6.2, we show that GTP is incapable of solving a simple task of classifying chiral 3D structures. However, the impact of the loss of antisymmetric interactions on real datasets remains unexplored.

### 3.3. Matrix Tensor Product

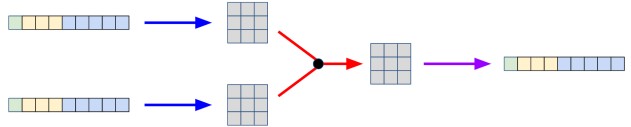

*Figure 3.* Schematic of the process in taking a matrix tensor product. We embed input irreps into a tensor product rep. We then interact using matrix multiplication before decomposing the resulting tensor product rep back into a direct sum of irreps.

We also analyze another interaction introduced in the new `e3x` framework in the `FusedTensor` class (Unke & Maennel, 2024; Maennel et al., 2024). The key idea is that a tensor product rep is a matrix, and we can interact two tensor product reps through matrix multiplication.

Matrix tensor products (MTP) first takes each input and embeds the irreps in a single large enough tensor product

rep using Clebsch-Gordan coefficients. After doing so, we can matrix multiply the tensor product reps. Finally, we can decompose the resulting tensor product rep back into irreps. Details are provided in Section E.3.

Similar to GTP, MTP only outputs one copy of each possible output irrep. Hence, the outputs of the same irrep type get weighted and summed together. However, in contrast to GTP, MTP is *not a symmetric operation* (because matrix multiplication is not commutative), so we can have antisymmetric tensor product terms. The selection rules are inherited from CGTP.

**Proposition 3.3** (Selection rule for MTP). Suppose we have irrep labels $(\ell_1, 1), (\ell_2, 1), (\ell_3, 1)$. We must have $\ell_a \leq \ell_b + \ell_c$ for all distinct choices of $a, b, c \in \{1, 2, 3\}$.

## 4. Summary of Asymptotic Runtimes

In Appendix E, we analyze the asymptotic runtimes, and in Appendix F, we analyze the expressivity of implementations of the various tensor product operations. The results are summarized in Table 1. For expressivity, we assume we are using the various tensor products to construct bilinearities from spaces $X, Y$ transforming as $(0 \oplus \ldots \oplus L)$ to space $Z$ transforming as $(0 \oplus \ldots \oplus 2L)$.

We would like to highlight the following findings on specific implementations. Details are found in Appendix E.

- The often cited $\mathcal{O}(L^6)$ runtime for CGTP does not leverage all the sparsity in the coefficients (Passaro & Zitnick, 2023). Using all sparsity, we can obtain $\mathcal{O}(L^5)$ runtime as noted by (Cobb et al.).

- We provide and analyze a **novel implementation** of GTP which represents spherical signals using a grid on the sphere as opposed to the original 2D Fourier basis (Luo et al., 2024). Our implementation has the **same asymptotics** and is 30% **faster in practice**. Futher, our spherical grid implementation unlocks the use of $S^2$ fast Fourier transforms (S2FFT) (Healy et al., 2003) algorithms which leads to an asymptotically faster version of GTP.

From this table, we can see that asymptotic speedups in the faster TPOs comes from a loss of expressivity. In particular, when normalizing for our expressivity measure, the only true asymptotic speedup comes from implementations leveraging the fast algorithm for spherical harmonic transforms.

## 5. Microbenchmarking Tensor Product Implementations

For the machine-learning practitioner, the fundamental question of which tensor product to choose for their network is

*Table 1.* Asymptotic runtimes and expressivity of various TPO implementations. Note that when normalized for expressivity, most TPOs have the same asymptotics as sparse CGTP. The only true speed up comes from a fast spherical transform algorithm by Healy et al. (2003).

| TPO | Expressivity | Runtime | Runtime / Expressivity |
|---|---|---|---|
| CGTP (Naive) | $\mathcal{O}(L^3)$ | $\mathcal{O}(L^6)$ | $\mathcal{O}(L^3)$ |
| CGTP (Sparse) | $\mathcal{O}(L^3)$ | $\mathcal{O}(L^5)$ | $\mathcal{O}(L^2)$ |
| GTP (Fourier) | $\mathcal{O}(L)$ | $\mathcal{O}(L^3)$ | $\mathcal{O}(L^2)$ |
| GTP (Grid) | $\mathcal{O}(L)$ | $\mathcal{O}(L^3)$ | $\mathcal{O}(L^2)$ |
| GTP (S2FFT) | $\mathcal{O}(L)$ | $\mathcal{O}(L^2 \log^2 L)$ | $\mathcal{O}(L \log^2 L)$ |
| MTP (Naive) | $\mathcal{O}(L)$ | $\mathcal{O}(L^4)$ | $\mathcal{O}(L^3)$ |
| MTP (Sparse) | $\mathcal{O}(L)$ | $\mathcal{O}(L^3)$ | $\mathcal{O}(L^2)$ |

often heavily dependent on the resulting wall-clock time of their training runs. In this section, we show that the practical runtimes of different tensor products is more *nuanced* than the theoretical guarantees of Table 1. We focus on the tensor products since these operations are usually the bottleneck in equivariant neural networks. We perform careful microbenchmarking on CPU (some details about the hardware here) and on GPU (specifically, the NVIDIA RTX A5500 and A100), and report hardware-agnostic FLOPs, hardware-dependent GPU utilization, and overall wall-clock time for each tensor product.

To the best of our knowledge, this is the most rigorous benchmarking of such tensor product operations performed till date.

### 5.1. Summary of Findings

We find that the Clebsch-Gordan Tensor Products incur low GPU utilization and higher wall-clock time when compared to the Gaunt Tensor Product and Matrix Tensor Product. However, if we normalize for expressivity (Appendix F), the wall-clock times for the Clebsch-Gordan Tensor Products are much lower.

Overall, our benchmarks highlight the need for carefully balancing expressivity, asymptotics, FLOPs and GPU utilization while designing and implementing tensor products.

### 5.2. Methodology

While wall-clock times of tensor products correlate well with downstream inference/training times, they depend on specific code implementations (eg. compiled vs uncompiled code, the use of custom kernels) or GPU compute capability (number of tensor cores, memory bandwidth). This makes it hard to extend benchmarking claims beyond the specific execution environment. While using FLOPS can help get around these limitations (Brehmer et al., 2024), they may not necessarily correspond to the actual compute needed for using equivariant operations given their poor GPU utilization. The difference in GPU utilization makes it particularly

challenging to do a fair comparison between different tensor products.

To test the analysis in Table 1, we implemented all of the tensor products in `e3nn-jax` built on top of the JAX (Bradbury et al., 2018) framework. The just-in-time compilation of JAX (via `jax.jit`) automatically fuses multiple operations to reduce memory transfer and kernel launch overhead. We also include an unweighted implementation of the matrix tensor product from `e3x` and a more GPU-friendly implementation of Clebsch-Gordan (Sparse) Algorithm 1 in our benchmarking suite. Through this we ensure that all tensor product implementations benefit equally from the various pattern-based fusion strategies implemented within JAX's XLA compiler (Sabne, 2020; Snider & Liang, 2023).

However, these fusion strategies and heuristics are still largely centered around dense linear algebra workloads over multi-dimensional arrays. This makes it harder to see out-of-the-box performance gains for operations that don't fit this paradigm (Barham & Isard, 2019). Thus, to analyze how efficiently a given tensor product is being executed on the GPU, we also report average GPU utilization.

**Benchmark Settings:** We define our benchmarking metrics by counting instructions executed within each kernel run, DRAM read and write memory accesses and throughput. All metrics were collected through NVIDIA's Nsight Compute toolkit. This approach has been successfully used in FourCastNet (Kurth et al., 2023) and takes inspiration from the Empirical Roofline Toolkit (Yang, 2020).

All of the experiments were performed on an NVIDIA A5500 with 24 GB of off-chip GPU memory. Our inputs are randomly generated and batch size refers to number of samples used at once. Benchmarks on other hardware (A100, CPU) can be found in Appendix M. Additional details about hardware counters can be found in Appendix L.

### 5.3. Results

**Asymptotics $\neq$ FLOPs**. The first trend we report is the discrepancy between asymptotics in Table 1 and compute FLOPs. While the GTPs have lower asymptotic complexity, they end up having comparable (or even higher) FLOPs than the CGTPs. This is due to different constant scaling factors causing TPOs with similar asymptotics to scale differently with $L_{\max}$.

**FLOPs $\neq$ Wall-clock time**. Next, we report discrepancy between FLOPs and wall-clock times. While GTP has higher FLOPs than the CGTPs, their wall-clock time is lower due to their high GPU utilization. For Gaunt (Fourier), our code does not leverage sparsity when transforming to a 2D Fourier basis, potentially causing the slowdown. We tried implementing a version that leveraged the sparsity through `jax.experimental.sparse` but it was slower than

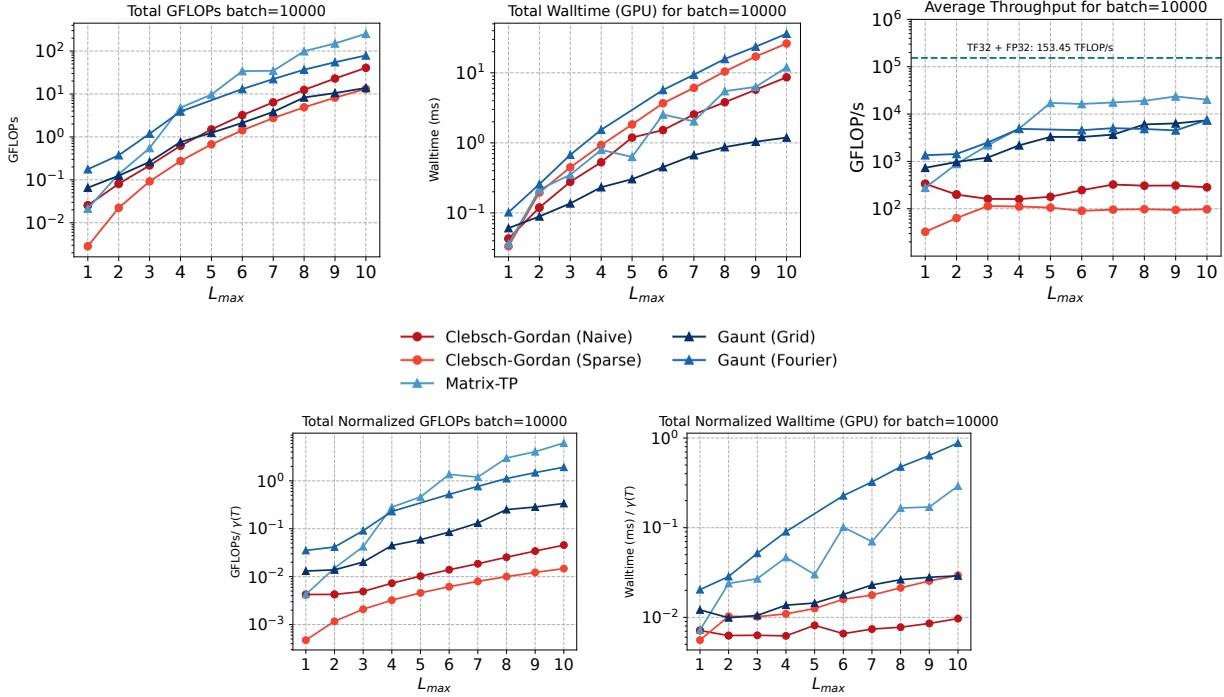

*Figure 4.* Top: Analysis of tensor products compute scaling on a RTX A5500 GPU: Total GFLOPs (Left), Total walltime (Middle), and Average throughput in GFLOPs/s (Right). Bottom: Analysis of tensor products compute scaling per path on RTX A5500 GPU: (Left) Total Walltime / Expressivity, (Right) Total GFLOPs / Expressivity. Batch refers to the number of tensor products performed in parallel.

the version we report suggesting that its not yet well optimized. We also report low GPU utilization for Clebsch-Gordan Tensor Product despite having a more GPU-friendly implementation Algorithm 1.

**GPU utilization $\neq$ Wall-clock time**. Incase of MTP, we show that high GPU utilization does not always necessarily lead to the lowest wall-clock time due to the high compute FLOPs.

**Wall-clock time $\neq$ Expressivity**. After normalizing against expressivity defined in Section 4, we find that the Clebsch-Gordan tensor products were the fastest both in terms of wall-clock time and FLOPs.

### 5.4. Limitations and Open Questions

Here, we have effectively only benchmarked the 'forward pass' of the tensor product operation in an equivariant neural network. In practice, depending on the dataset and the task, it may turn out the 'optimal' tensor product here may not result in significant performance gains. For example, the lack of expressivity in some of these tensor products may be captured in other operations of the neural network, or may not be important to the task at hand.

Since Nsight Compute does multiple replays on a single kernel in order to gather performance metrics, it can take

days to profile all the kernels being executed in a single model inference call. This is the biggest bottleneck when profiling model runs.

By reporting the GPU utilization and FLOPs of these tensor products, we hope to highlight the performance gains relative to the device peak that are still left on the table. Recent works (NVIDIA, 2024; Bharadwaj et al., 2025; Firoz et al., 2025; Tan et al., 2025) show this is a promising direction.

## 6. Experiments

### 6.1. Force and Energy Prediction on Organic Molecules

Luo et al. (2024) demonstrates the speedups obtained by GTP in a real-world application by replacing the symmetrized tensor product in the MACE (Batatia et al., 2022) model (a popular machine learning interatomic potential) with GTP. We redo their experiments on the 3BPA (Kovács et al., 2021) and revised MD17 (Christensen & von Lilienfeld, 2020) datasets with our S2Grid implementation of GTP. Our implementation is a *drop-in replacement* for the original GTP implementation using Fourier transforms. While significantly simpler due to the use of pre-existing primitives in e3nn, we find that our GTP implementation using S2Grid is also $\approx 30\%$ faster in practice, as shown in Figure 5.

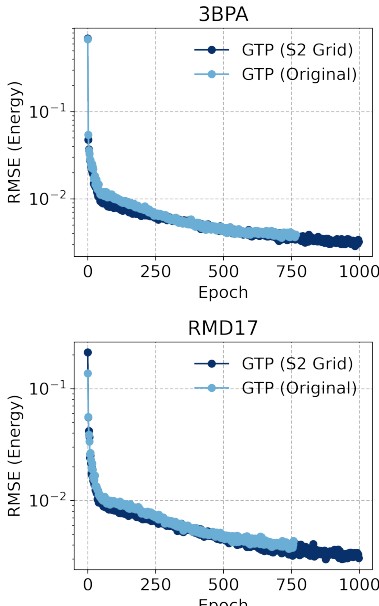

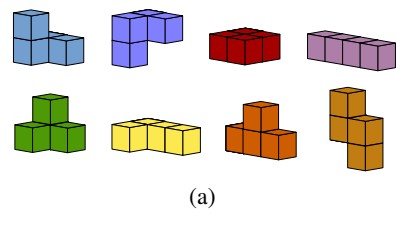

(a)

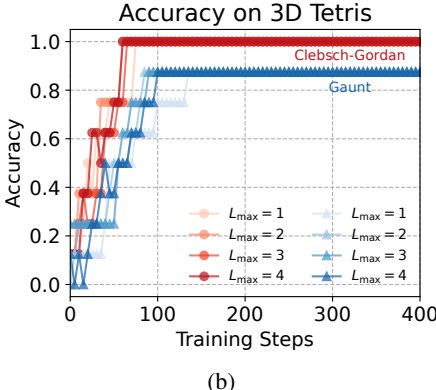

(b)

*Figure 5.* Training the MACE interatomic force field with the original GTP (Luo et al., 2024) and our faster implementation GTP (Grid) for the same number of GPU hours on the 3BPA and revised MD17 datasets.

*Figure 6.* (a) The 8 different 3D Tetris pieces, with the first two pieces being mirror images of each other. (b) Training curves of networks trained with different tensor products on the 3D Tetris task. The maximum $L$ is varied from 1 to 4. All of the CGTP networks attain $100\%$ accuracy while none of the GTP networks do.

## 6.2. Classifying 3D Tetris Pieces

We consider a simple task of classifying 8 different 3D Tetris-like pieces, shown in 6a. Note that the first two pieces are non-superimposable mirror reflections of each other; they are *chiral*. Given a randomly oriented 3D structure, the network needs to predict which of the 8 tetris pieces it corresponds to.

We use a simple message-passing neural network, described in Appendix I, using either the Gaunt and Clebsch-Gordan tensor products. Our network architecture is almost identical to that of NequIP (Batzner et al., 2022).

As shown in Figure 6, the network is very easily able to solve this task with CGTP, but the same network parametrized with GTP is unable to distinguish between the two chiral pieces. Adding more channels or incorporating the pseudo-spherical harmonics (which have the opposite parity of the spherical harmonics under reflection) did not help. The fundamental failure is the inability to create the $1e$ term via $1o \otimes 1o \rightarrow 1e$ because this is the cross product, an antisymmetric operation. Indeed, there is no way to create a pseudoscalar using the GTP in this setting.

## 7. Conclusion

This work was inspired by the observation that specific antisymmetric paths were missing in GTP and that paths which result in the same output irrep type are merged together.

While there is much focus on improving how runtimes scale with $L$, there has been little work analyzing how expressivity is affected. Hence we provided the first systematic analysis of the distinctions between different $O(3)$ equivariant TPOs that exist in the literature.

On the theoretical side, we clarify that many new TPOs are not formally tensor products. We then introduced a measure of expressivity and interactability based on how TPOs are used in practice. We applied this to CGTP, GTP, and MTP analyzing their asymptotic runtimes, expressivities, and interactability. Interactability lets us formalize the intuition that GTP is unable to represent antisymmetric interactions. In addition, we see asymptotic speedups in runtime typically come from losses in expressivity.

For implementations, we highlight our novel and simpler implementation of GTP which directly uses a spherical grid. This implementation has the same asymptotic cost as the original GTP and perfroms 20% faster in practice. Further, we identify that leveraging fast spherical harmonic transforms can allow faster asymptotic runtimes. Finally, we clarify that the commonly cited $\mathcal{O}(L^6)$ runtime for CGTP does not use all sparsity in the Clebsch-Gordan coefficients and we can in fact achieve $\mathcal{O}(L^5)$ runtime (Cobb et al.).

We then provide the first microbenchmarks of implementations of CGTP, GTP, and MTP. We observe asymptotic

gains do not always correspond to performance gains in practice. In fact, CGTP leveraging sparsity has the slowest walltime despite having the fewest FLOPs. As expected, we observe better walltime performance for GTP compared with CGTP. In addition, our grid implementation of GTP performs better than the original 2D Fourier implementation. However, normalizing for our expressivity measure, we see that standard CGTP has the best performance, highlighting these new TPOs do not truly speed up tensor products but rather remove degrees of freedom.

Finally, we replicated the experiments originally performed in Luo et al. (2024). We directly replaced their GTP implementation with our grid implementation and observe a 30% speed increase. Additionally, we demonstrate the lack of antisymmetric operations means classifying 3D tetris pieces using solely GTP is impossible.

We conclude that while it is important to improve runtimes, it is equally important to properly analyze where the savings come from and the impact of these new operations both in theory and in practice. We believe there are many opportunities for creative design of new TPOs. We hope this work can provide a foundation for analyzing these operations.

## Acknowledgments

YuQing Xie, Ameya Daigavane, and Mit Kotak were supported by the NSF Graduate Research Fellowship program under Grant No. DGE-1745302. Tess Smidt was supported by DOE ICDI grant DE-SC0022215. Mit Kotak and Tess Smidt would like to additionally acknowledge support from the National Science Foundation under Cooperative Agreement PHY-2019786 (The NSF AI Institute for Artificial Intelligence and Fundamental Interactions).

## Impact Statement

This paper presents work whose goal is to advance the field of Machine Learning. There are many potential societal consequences of our work, none which we feel must be specifically highlighted here.

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

## A. Notation

Here, we present the notation we use throughout this paper and the typical variable names.

*Table 2.* Notation used throughout this paper

| | |
|---|---|
| $SO(n)$ | Group of rotations in $n$-dimensional space |
| $O(n)$ | Group of rotations and inversion in $n$-dimensional space |
| $S^2$ | The 2-sphere, surface defined by $x^2 + y^2 + z^2 = 1$ |
| $Y_\ell^m$ | Spherical harmonic function of degree $\ell$ and order $m$ |
| $\mathbf{Y}_\ell$ | The collection of spherical harmonic functions of degree $\ell$ for all orders $m$ |
| $\oplus$ | Denotes a direct sum |
| $\otimes$ | Denotes a tensor product |
| $\times$ | Denotes a Cartesian product of spaces and also denotes cross products |
| $\mathcal{O}$ | Big O notation |
| ToSphere | Function which takes in spherical harmonic coefficients consisting of single copies of irrep up to some cutoff $L$ and converts it into a spherical signal $f : S^2 \to \mathbb{R}$ |
| FromSphere | Function takes in a spherical signal $f : S^2 \to \mathbb{R}$ and converts it to spherical harmonic coefficients consisting of single copies of irrep up to some cutoff $L$ |

*Table 3.* Commonly used meanings of symbols

| | |
|---|---|
| $G$ | Denotes a group |
| $\rho(g)$ | Representation of a group |
| $A$ | First input space of a constructed bilinearity |
| $B$ | Second input space of a constructed bilinearity |
| $C$ | Output space of a constructed bilinearity |
| $V$ | First input space of a tensor product operation (fixed equivariant bilinearity) |
| $W$ | Second input space of a tensor product operation (fixed equivariant bilinearity) |
| $V$ | Output space of a tensor product operation (fixed equivariant bilinearity) |
| $T$ | Tensor product operation (fixed equivariant bilinearity $V \times W \to Z$) |
| $\ell$ | Typically used to denote irrep type for $SO(3)$. For spherical signals, used instead to denote spherical harmonic degree which naturally indexes multiplicities of irrep types for VSTP/ISTPs |
| $c$ | Indexes multiplicities of an irrep type in given space (channel) |
| $j$ | Used instead of $\ell$ to denote irrep type for VSTP and general ISTPs |
| $s$ | Denotes irrep type of spherical signal (ie. our signal is a map $f : S^2 \to \mathbb{R}^{2s+1}$) |

## B. Irreducible Representations of $E(3)$

A representation $\rho$ of a group $G$ maps each group element $g$ to a bijective linear transformation $\rho(g) \in \mathrm{GL}(V)$, where $V$ is some vector space. Representations must preserve the group multiplication property:

$$\rho(g \cdot h) = \rho(g) \circ \rho(h) \quad \forall g, h \in G \tag{4}$$

Thus, the representation $\rho$ defines a group action on a vector space $V$. The dimension of the representation $\rho$ is simply defined as the dimension of the vector space $V$.

There may be subspaces $W \subset V$ which are left invariant under actions of $\rho(g)$ for all $g \in G$. If this is the case, then restricting to $W$ also gives a representation $\rho|_W(g) \in \mathrm{GL}(W)$. If there is no nontrivial $W$, then we say the representation $\rho$ is an irreducible representation (irrep).

To build $E(3)$-equivariant neural networks, the irreducible representations of $E(3)$ play a key role. Because $E(3)$ is not a compact group, the usual approach has been to consider irreducible representations of the group $SO(3)$ of 3D rotations, and

compose them with the representation in which translations act as the identity:

$$\rho(R, T) = \rho'(R) \tag{5}$$

This is why translations are often handled in $E(3)$-equivariant neural networks by centering the system or only using relative vectors.

The 'scalar' representation $\rho_{\text{scalar}}$ representation of $SO(3)$ is defined as:

$$\rho_{\text{scalar}}(R) = \text{id} \quad \forall R \in SO(3) \tag{6}$$

and is of dimension 1 over $V = \mathbb{R}$. Elements of $\mathbb{R}$ are unchanged by any rotation $R$. We call such elements 'scalars' to indicate that they transform under the 'scalar' representation of $SO(3)$. An example of a 'scalar' element could be mass of an object, which does not change under rotation of coordinate frames.

Let $T(R) \in \mathbb{R}^{3 \times 3}$ be the rotation matrix corresponding to a rotation $R \in SO(3)$. Then, the 'vector' representation of $SO(3)$ is defined as:

$$\rho_{\text{vector}}(R) = T(R) \quad \forall R \in SO(3) \tag{7}$$

and is of dimension 3 over $V = \mathbb{R}^3$. The name arises from the way vectors in $\mathbb{R}^3$ transform under a rotation of the coordinate frame. We call such elements 'vectors' to indicate that they transform under the 'vector' representation of $SO(3)$. For example, the velocity and position of an object in a certain coordinate frame are 'vectors'.

Weyl's theorem for the Lie group $SO(3)$ states that all finite-dimensional representations of $SO(3)$ are equivalent to direct sums of irreducible representations. The irreducible representations of $SO(3)$ are indexed by an integer $\ell \geq 0$, with dimension $2\ell + 1$. $\ell = 0$ corresponds to the 'scalar' representation, while $\ell = 1$ corresponds to the 'vector' representation above. We will often use $m$, where $-\ell \leq m \leq \ell$, to index of each of the $2\ell + 1$ components.

We say that a quantity $\mathbf{x} \in \mathbb{R}^{2\ell+1}$ is a $\ell$ irrep, if it transforms as the irreducible representation ('irrep') of $SO(3)$ indexed by $\ell$. If $\mathbf{x}_1$ is a $\ell_1$ irrep and $\mathbf{x}_2$ is an $\ell_2$ irrep, we say that $(\mathbf{x}_1, \mathbf{x}_2)$ is a direct sum of $\ell_1$ and $\ell_2$ irreps, which we call a $(\ell_1, \ell_2)$ 'rep'. Weyl's theorem states that all reps are a direct sum of $\ell_i$ irreps, possibly with repeats over $\ell_i$: $\mathbf{x} = \oplus_{\ell_i} \mathbf{x}^{(\ell_i)}$. The multiplicity of an irrep in a rep is exactly the number of repeats.

An important lemma for constructing equivariant linear layer is Schur's lemma (Dresselhaus et al., 2007).

**Lemma B.1** (Schur's Lemma). Suppose $V_1, V_2$ are irreps of a Lie group over any algebraically closed field (such as $SO(3)$). Let $\phi : V_1 \to V_2$ be an equivariant linear map. Then $\phi$ is either 0 or an isomorphism. Further, if $V_1 = V_2$ then $\phi$ is a multiple of identity. Finally, for any two $\phi_1, \phi_2 : V_1 \to V_2$ we must have $\phi_1 = \lambda \phi_2$.

This tells us that to construct equivariant linear layers between reps written as a direct sum of irreps, we can only have weights between input and output irreps of the same type and that those weights must be tied together so they give multiples of the identity transformation.

## C. Spherical Harmonics

The spherical harmonics are intimately connected to the representations of $SO(3)$ and play a key role in the Gaunt tensor product.

We define the spherical coordinates $(r, \theta, \varphi)$ as:

$$\begin{bmatrix} x \\ y \\ z \end{bmatrix} = \begin{bmatrix} r \sin\theta \cos\varphi \\ r \sin\theta \sin\varphi \\ r \cos\theta \end{bmatrix} \tag{8}$$

for $\theta \in [0, \pi), \varphi \in [0, 2\pi)$.

The spherical harmonics $Y_{\ell,m}$ are a set of functions $S^2 \to \mathbb{R}$ indexed by $(\ell, m)$, where again $\ell \geq 0, -\ell \leq m \leq \ell$. Here, $S^2 = \{(r, \theta, \phi) \mid r = 1\}$ denotes the unit sphere.

Indeed, as suggested by the notation, the spherical harmonics are closely related to the irreducible representations of $SO(3)$. Let $Y_\ell$ be the concatenation of all $Y_{\ell,m}$ over all $m$ for a given $\ell$:

$$Y_\ell(\theta, \phi) = \begin{bmatrix} Y_{\ell,-\ell}(\theta, \phi) \\ Y_{\ell,-\ell+1}(\theta, \phi) \\ \dots \\ Y_{\ell,\ell}(\theta, \phi) \end{bmatrix} \tag{9}$$

When we transform the inputs to $Y_\ell(\theta, \phi)$, the output transforms as a $\ell$ irrep.

The spherical harmonics satisfy orthogonality conditions:

$$\int_{S^2} Y_{\ell_1,m_1} \cdot Y_{\ell_2,m_2} \, dS^2 = \delta_{\ell_1 \ell_2} \delta_{m_1 m_2} \tag{10}$$

where:

$$\int_{S^2} f \cdot g \, dS^2 = \int_{\theta=0}^{\pi} \int_{\varphi=0}^{2\pi} f(\theta, \varphi) g(\theta, \varphi) \sin \theta d\theta d\varphi \tag{11}$$

The orthogonality property allows us to treat the spherical harmonics as a basis for functions on $S^2$. We can linearly combine the spherical harmonics using irreps to approximate arbitrary functions on the sphere. Given a $(0, 1, \dots, L)$ rep $\mathbf{x} = (\mathbf{x}^{(0)}, \mathbf{x}^{(1)}, \dots, \mathbf{x}^{(L)})$, we can associate the function $f_{\mathbf{x}} : S^2 \to \mathbb{R}$ as:

$$f_{\mathbf{x}}(\theta, \varphi) = \sum_{\ell=0}^{L} \sum_{m=-\ell}^{\ell} \mathbf{x}_m^{(\ell)} Y_{\ell,m}(\theta, \varphi) \tag{12}$$

The function $f_{\mathbf{x}}$ is uniquely determined by $\mathbf{x}$. In particular, by the orthogonality of the spherical harmonics (Equation 10), we can recover the $\mathbf{x}_m^{(\ell)}$ component:

$$\mathbf{x}_m^{(\ell)} = \int_{S^2} f_{\mathbf{x}} \cdot Y_{\ell,m} \, dS^2 \tag{13}$$

Thus, we can define the operations ToSphere and FromSphere:

$$\mathbf{x} \xrightarrow{\text{ToSphere}} f_{\mathbf{x}} \xrightarrow{\text{FromSphere}} \mathbf{x} \tag{14}$$

## D. Clebsch-Gordan Tensor Product Details

The most natural map is $V \times W \to V \otimes W$ constructed by taking an outer product of the inputs. If the inputs are explicitly written as a direct sum of irreps, we can write the tensor product as

$$\mathbf{x} \otimes \mathbf{y} = \bigoplus_{\substack{\mathbf{x}^{(\ell_1)} \in \mathbf{x} \\ \mathbf{y}^{(\ell_2)} \in \mathbf{y}}} (\mathbf{x}^{(\ell_1)} \otimes \mathbf{y}^{(\ell_2)}) \tag{15}$$

a new basis which is the sum of tensor product reps.

The key idea of a Clebsch-Gordan tensor product is we can explicitly reduce the tensor product reps back into a direct sum of irreps with a change of basis. This change of basis is the definition of the Clebsch-Gordan coefficients, giving us

$$\mathbf{x}^{(\ell_1)} \otimes \mathbf{y}^{(\ell_2)} = \bigoplus_{\ell_3} (\mathbf{x}^{(\ell_1)} \otimes_{\text{CG}} \mathbf{y}^{(\ell_2)})^{(\ell_3)} \tag{16}$$

where

$$
(\mathbf{x}^{(\ell_1)} \otimes_{\text{CG}} \mathbf{y}^{(\ell_2)})_{m_3}^{(\ell_3)}
$$
$$
= \sum_{m_1=-\ell_1}^{\ell_1} \sum_{m_2=-\ell_2}^{\ell_2} C_{\ell_1,m_1,\ell_2,m_2}^{\ell_3,m_3} \mathbf{x}_{m_1}^{(\ell_1)} \mathbf{y}_{m_2}^{(\ell_2)}. \tag{17}
$$

Therefore the original tensor product can also be rewritten as a direct sum of irreps. This defines the Clebsch-Gordan tensor product (CGTP)

$$\mathbf{x} \otimes_{\mathrm{CG}} \mathbf{y} = \bigoplus_{\substack{\mathbf{x}^{(\ell_1)} \in \mathbf{x} \\ \mathbf{y}^{(\ell_2)} \in \mathbf{y}}} (\mathbf{x}^{(\ell_1)} \otimes_{\mathrm{CG}} \mathbf{y}^{(\ell_2)}). \tag{18}$$

Note that full CGTP is really just a change of basis from a sum of tensor product reps to a sum of irreps. Hence, we **do not lose any information**.

## E. Runtime Analysis

Here, we provide a detailed asymptotic analysis of runtimes for different tensor products. We consider 3 different settings.

- **Single Input, Single Output (SISO)**:

  Here we are computing only one path $[\ell_1, \ell_2, \ell_3]$ where $\ell_i \in \mathcal{O}(L)$.

  $$\ell_1 \times \ell_2 \to \ell_3$$

- **Single Input, Multiple Output (SIMO)**:

  Here we fix $\ell_1, \ell_3$ but allow all possible irreps generated by the respective tensor products.

  $$\ell_1 \times \ell_2 \to Z$$

- **Multiple Input, Multiple Output (MIMO)**:

  Here we only bound the $L$ that the tensor products use but allow full capacity for the input and output irreps. In the case of CGTP, we can have an arbitrary number of copies of each irrep but we assume we only use single copies of each irrep in the input.

  $$Z \times W \to Z$$

In the SISO and SIMO settings, the asymptotic runtimes of different tensor products are directly comparable. However, in the MIMO setting, we lose expressivity in some tensor products. This is discussed more in Appendix F. Note the MIMO setting is what one would typically want to use in practice. Hence the runtimes reported in Table 1 are for the MIMO setting.

### E.1. Clebsch-Gordan Tensor Product

The tensor product operation is defined as:

$$(\mathbf{x}^{(\ell_1)} \otimes_{\mathrm{CG}} \mathbf{x}^{(\ell_2)})_{m_3}^{(\ell_3)} = \sum_{m_1=-l_1}^{l_1} \sum_{m_2=-l_2}^{l_2} C_{\ell_1,m_1,\ell_2,m_2}^{(\ell_3,m_3)} \mathbf{x}_{m_1}^{(\ell_1)} \mathbf{x}_{m_2}^{(\ell_2)} \tag{19}$$

where $C$ denotes the Clebsch-Gordan (CG) coefficients which can be precomputed.

#### E.1.1. NAIVE RUNTIME

Let $L = \max(\ell_1, \ell_2, \ell_3)$. From Equation 19, for each $m_3$, we would need to sum over $m_1, m_2$ which range from $-\ell_1$ to $\ell_1$ and $-\ell_2$ to $\ell_2$ respectively. Hence, we expect $\mathcal{O}(L^2)$ operations. To compute the values for all $m$ which range from $-\ell_3$ to $\ell_3$, we see that computing a single $\ell_1 \otimes \ell_2 \to \ell_3$ tensor product requires $\mathcal{O}(L^3)$ operations.

#### E.1.2. OPTIMIZED RUNTIME WITH SPARSITY

However, the CG coefficients are sparse. In the complex basis for the irreps, $C_{\ell_1,m_1,\ell_2,m_2}^{(\ell_3,m)}$ is nonzero only if $m_1 + m_2 = m_3$. Transforming to the real basis for the irreps, this condition becomes $\pm m_1 \pm m_2 = m_3$. In either case for a fixed $m_1$ and $m_3$, we only ever need to sum over a constant number of $m_2$'s rather than $\mathcal{O}(L)$ of them as naively expected. Therefore an implementation taking this sparsity into account gives us a runtime of $\mathcal{O}(L^2)$. This optimization was noted in Cobb et al..

### E.2. Gaunt Tensor Product

The Gaunt Tensor Product (GTP) is based on the decomposition of a product of spherical harmonic functions back into spherical harmonics (Luo et al., 2024). In particular, suppose one of our inputs $\mathbf{x}^{(\ell_1)}$ transforms as a direct sum of irreps up to some cutoff $L$ (ie. $\ell_1$ ranges from $0, \ldots, L$). We can view these irreps as coefficients of spherical harmonics which gives a spherical signal $F_1(\theta, \varphi) = \sum_{\ell_1, m_1} \mathbf{x}_{m_1}^{(\ell_1)} Y_{\ell_1, m_1}(\theta, \varphi)$. We similarly construct $F_2(\theta, \varphi) = \sum_{\ell_2, m_2} \mathbf{x}_{m_2}^{(\ell_2)} Y_{\ell_2, m_2}(\theta, \varphi)$.

Taking the product of these spherical signals gives a new signal $F_3(\theta, \varphi) = F_1(\theta, \varphi) F_2(\theta, \varphi)$. This new signal can be decomposed into spherical harmonics which we use to define the GTP. This results in

$$F_3(\theta, \varphi) = \sum_{\ell_3, m_3} (\mathbf{x}^{(\ell_1)} \otimes_{\mathrm{GTP}} \mathbf{x}^{(\ell_2)})_{m_3}^{(\ell_3)} Y_{\ell_3, m_3}(\theta, \varphi). \tag{20}$$

#### E.2.1. 2D FOURIER BASIS

Luo et al. (2024) describe an implementation which decomposes spherical harmonics into a 2D Fourier basis in their original paper introducing GTP. This also turns out to be the same implementation in Xin et al. (2021). We describe their procedure here.

Note that for any $\ell \le L$ we can always write the spherical harmonics in the 2D Fourier basis:

$$Y_{\ell,m}(\theta, \varphi) = \sum_{-L \le u,v \le L} y_{u,v}^{\ell,m} e^{i(u\theta + v\varphi)} \tag{21}$$

for some coefficients $y_{u,v}^{\ell,m}$.

Hence, any signal $\mathbf{x}_m^{(\ell)}$ can be encoded as

$$F_1(\theta, \varphi) = \sum_{\ell=0}^{L} \sum_{m=-\ell}^{\ell} \sum_{-L \le u,v \le L} \mathbf{x}_m^{(\ell)} y_{u,v}^{\ell,m} e^{i(u\theta + v\varphi)} = \sum_{-L \le u,v \le L} \left( \sum_{\ell=0}^{L} \sum_{m=-\ell}^{\ell} \mathbf{x}_m^{(\ell)} y_{u,v}^{\ell,m} \right) e^{i(u\theta + v\varphi)}. \tag{22}$$

We identify the encoding

$$\mathbf{x}_{u,v} = \sum_{\ell=0}^{L} \sum_{m=-\ell}^{\ell} \mathbf{x}_m^{(\ell)} y_{u,v}^{\ell,m}. \tag{23}$$

One can observe that the $y_{u,v}^{\ell,m}$ are sparse and only nonzero when $m = \pm v$. Therefore, finding $\mathbf{x}_{u,v}$ if we have a set of irreps is $\mathcal{O}(L)$ and it is $O(1)$ if we only want one irrep. Because there are $\mathcal{O}(L^2)$ possible values for $u, v$, encoding into the 2D Fourier is $\mathcal{O}(L^3)$ if we encode all irreps up to $L$ or $\mathcal{O}(L^2)$ if encoding a single irrep.

For 2 functions of $\theta, \varphi$ encoded using a 2D Fourier basis $\mathbf{x}_{u,v}^1, \mathbf{x}_{u,v}^2$, we can compute their product using a standard 2D FFT in $\mathcal{O}(L^2 \log L)$ time. This gives some output encoded as $\mathbf{y}_{u,v}$ where now $u, v$ range from $-2L, \ldots, 2L$ to capture all information.

Finally, we decode the resulting function in the 2D Fourier basis back into a spherical harmonic basis to extract the output irreps. Suppose $-L \le u, v \le L$. We can always write

$$e^{i(u\theta + v\varphi)} = F_{u,v}^{\perp}(\theta, \varphi) + \sum_{\ell=0}^{L} \sum_{m=-\ell}^{\ell} z_{u,v}^{\ell,m} Y_{\ell,m}(\theta, \varphi) \tag{24}$$

where $F_{u,v}^{\perp}(\theta, \varphi)$ is some function in the space orthogonal to that spanned by the spherical harmonics. By construction, our output signal is always in the space spanned by the spherical harmonics so the orthogonal parts cancel. Hence we can write

$$\sum_{-2L \le u,v \le 2L} \mathbf{y}_{u,v} e^{i(u\theta + v\varphi)} = \sum_{-2L \le u,v \le 2L} \mathbf{y}_{u,v} \sum_{\ell=0}^{L} \sum_{m=-\ell}^{\ell} z_{u,v}^{\ell,m} Y_{\ell,m}(\theta, \varphi) \tag{25}$$

$$= \sum_{\ell=0}^{L} \sum_{m=-\ell}^{\ell} \left( \sum_{-2L \le u,v \le 2L} \mathbf{y}_{u,v} z_{u,v}^{\ell,m} \right) Y_{\ell,m}(\theta, \varphi) \tag{26}$$

Hence we identify:

$$\mathbf{y}_m^\ell = \sum_{-2L \le u,v \le 2L} \mathbf{y}_{u,v} z_{u,v}^{\ell,m}. \tag{27}$$

Once again, we can note that $z_{u,v}^{\ell,m}$ must be sparse and is only nonzero when $v = \pm m$. Hence, evaluating the above takes $\mathcal{O}(L)$ time since we sum over $\mathcal{O}(L)$ values of $u$ paired with constant number of $v$'s. If we only extract one irrep, then we range over $\mathcal{O}(L)$ values of $m$ giving $\mathcal{O}(L^2)$ runtime. If we extract all irreps up to $2L$ this becomes $\mathcal{O}(L^3)$.

### E.2.2. GRID TENSOR PRODUCT

Rather than use a 2D Fourier basis, we can instead represent the signal by directly giving its value for a set of points on the sphere. Quadrature on the sphere is a well-studied topic (Beentjes, 2015; Lebedev, 1976); in general, $\mathcal{O}(L^2)$ points are needed to exactly integrate spherical harmonics upto degree $L$ (McLaren, 1963). For this section, consider a product grid on the sphere formed by the Cartesian product of two 1D grids for $\theta$ and $\varphi$ with $\mathcal{O}(L)$ points each, for a total of $\mathcal{O}(L^2)$ points.

We can write:

$$F_1(\theta_j, \varphi_k) = \sum_{\ell=0}^{L} \sum_{m=-\ell}^{\ell} \mathbf{x}_m^{(\ell)} Y_{\ell,m}(\theta_j, \varphi_k) = \sum_{\ell=0}^{L} \sum_{m=-\ell}^{\ell} \mathbf{x}_m^{(\ell)} N_{\ell,m} P_\ell^m(\cos(\theta_j)) cs_m(\varphi_k) \tag{28}$$

where $N_{\ell,m}$ is some normalization factor, $P_\ell^m$ are the associated Legendre polynomials, and

$$cs_m(\varphi) = \begin{cases} \sin(|m|\varphi) & m < 0 \\ 1 & m = 0 \\ \cos(m\varphi) & m > 0 \end{cases} . \tag{29}$$

We note that we can first evaluate

$$g_m(\theta_j) = \sum_{\ell=0}^{L} \mathbf{x}_m^{(\ell)} N_{\ell,m} P_\ell^m(\cos(\theta_j)) \tag{30}$$

where we set $P_\ell^m = 0$ if $m > \ell$. If we have a set of irreps up to $L$ then we do the summation and this takes $\mathcal{O}(L)$ time. If we only have one irrep to encode then this takes $O(1)$ time. But we also have $\mathcal{O}(L)$ values of $\theta_j$ on the grid and $\mathcal{O}(L)$ values of $m$ to evaluate. This gives $\mathcal{O}(L^3)$ runtime to encode onto the grid for irreps up to $L$ and $\mathcal{O}(L^2)$ for a single irrep. Finally evaluating

$$F_1(\theta_j, \varphi_k) = \sum_{m=-\ell}^{\ell} g_m(\theta_j) cs_m(\varphi_k) \tag{31}$$

for a set of $\varphi_k$ can be done through a FFT in $\mathcal{O}(L \log L)$ time for each $\theta_j$ giving $\mathcal{O}(L^2 \log L)$ total. Hence we see encoding onto the sphere takes $\mathcal{O}(L^3)$ time for irreps up to $L$ and $\mathcal{O}(L^2 \log L)$ time for a single irrep.

For the multiplication of signals, we just have elementwise multiplication $F_3(\theta_k, \varphi_k) = F_1(\theta_k, \varphi_k) \cdot F_2(\theta_k, \varphi_k)$. Since there are $\mathcal{O}(L^2)$ grid points this takes $\mathcal{O}(L^2)$ time.

Finally, we decode the signal back into irreps. To do so we use the fact that

$$\mathbf{f}_m^{(\ell)} = \sum_{j,k} a_j F(\theta_j, \varphi_k) Y_{\ell,m}(\theta_j, \varphi_k) \tag{32}$$

for some coefficients $a_j$. This is essentially performing numerical integration of our signal against a spherical harmonic. Once again using the factorization of the spherical harmonics we get

$$\mathbf{f}_m^{(\ell)} = \sum_j \left( \sum_k F(\theta_j, \varphi_k) cs_m(\varphi_k) \right) a_j N_{\ell,m} P_\ell^m(\cos(\theta_j)). \tag{33}$$

The inner sum in parentheses can be computed in $\mathcal{O}(L)$ time and we need to compute it for $\mathcal{O}(L^2)$ values of $\theta_j, m$ pairs giving a runtime of $\mathcal{O}(L^3)$. Of course, we note that $cs$ really is just sines and cosines so alternatively we can use FFT which takes $\mathcal{O}(L^2 \log L)$ total. Computing the outer sum takes $\mathcal{O}(L)$ since we sum over $\mathcal{O}(L)$ values of $j$. For a single irrep there are $\mathcal{O}(L)$ values of $j$ giving $\mathcal{O}(L^2)$ for the outer sum. For irreps up to $\ell$ there are $\mathcal{O}(L^2)$ pairs of $\ell, m$ giving $\mathcal{O}(L^3)$ runtime for the outer sum. In total, we see going from the grid to the coefficients takes $\mathcal{O}(L^2 \log L)$ for a single irrep and $\mathcal{O}(L^3)$ for all irreps.

However, it turns out that the associated Legendre polynomials have recurrence properties which can be exploited to make transforming a set of irreps up to $L$ to the grid and a set of irreps up to $L$ back from the grid asymptotically more efficient (Healy et al., 2003). The runtime for this algorithm which we will call S2FFT is $\mathcal{O}(L^2 \log^2 L)$.

## E.3. Matrix Tensor Product

Here we describe and analyze the time complexity of matrix tensor product. Let $L_1, L_2$ be the max $\ell$'s of the inputs and $L_3$ be the max $\ell$ of the outputs. We pick some $\tilde{\ell} = \lceil \max(L_1, L_2, L_3)/2 \rceil$ so that $\tilde{\ell} \otimes \tilde{\ell}$ when decomposed into irreps can contain all irreps of the inputs and outputs. Note in principle we could always choose larger $\tilde{\ell}$.

In the following runtime analysis, we assume $L_1 = L_2 = L$, $\tilde{l} = L$, and $L_3 = 2L$.

### E.3.1. NAIVE RUNTIME

The first step of MTP is to convert our input irreps into a tensor product rep using Clebsch-Gordan coefficients as

$$\mathbf{X}_{m_1,m_2}^{(\ell)} = \sum_{m_3=-\ell}^{\ell} C_{\tilde{\ell},m_1,\tilde{\ell},m_2}^{\ell_3,m_3} \mathbf{x}_{m_3}^{(\ell)} \tag{34}$$

$$\mathbf{Y}_{m_1,m_2}^{(\ell)} = \sum_{m_3=-\ell}^{\ell} C_{\tilde{\ell},m_1,\tilde{\ell},m_2}^{\ell_3,m_3} \mathbf{y}_{m_3}^{(\ell)}. \tag{35}$$

Naively we sum over $\mathcal{O}(L)$ values of $m_3$ and need to do the computation for $\mathcal{O}(L^2)$ possible pairs of $m_1, m_2$. This gives $\mathcal{O}(L^3)$ naive runtime for converting a single irrep into a tensor product rep. To do so for all irreps up to $L$ the takes $\mathcal{O}(L^4)$ time.

We can then sum over tensor product reps to create

$$\mathbf{X} = \sum_{\ell} \mathbf{X}^{(\ell)} \qquad \mathbf{Y} = \sum_{\ell} \mathbf{Y}^{(\ell)}. \tag{36}$$

There are $\mathcal{O}(L)$ matrices to sum over if we have irreps up to $L$. Summing matrices takes $\mathcal{O}(L^2)$ time since our matrices are size $\mathcal{O}(L) \times \mathcal{O}(L)$. Hence, this takes $\mathcal{O}(L^3)$ time if we have irreps up to $L$. If we have a single irrep then we do not need to do anything.

We then multiply the matrices giving $\mathbf{Z} = \mathbf{XY}$. Using the naive matrix multiplication algorithm requires $\mathcal{O}(L^3)$ runtime.

Finally we can use Clebsch-Gordan to extract individual irreps giving

$$(\mathbf{x} \otimes_{\text{FTP}} \mathbf{y})_{m_3}^{(\ell_3)} = \sum_{m_1=-\ell_1}^{\ell_1} \sum_{m_2=-\ell_2}^{\ell_2} C_{\ell_1,m_1,\ell_2,m_2}^{(\ell_3,m_3)} \mathbf{Z}_{m_1,m_2}. \tag{37}$$

Again, naively we sum over $\mathcal{O}(L^2)$ pairs of $m_1, m_2$ and need to evaluate $\mathcal{O}(L)$ values of $m_3$ for $\mathcal{O}(L^3)$ conversion for single irrep. If we want all irreps up to $2L$ then we need $\mathcal{O}(L^4)$.

### E.3.2. OPTIMIZED RUNTIME WITH SPARSITY

Similar to the CGTP, we can take sparsity of the Clebsch-Gordan coefficients into account. We have nonzero values only if $\pm m_1 \pm m_2 = m_3$. Hence in the encoding step, for fixed $m_1, m_2$ we only need to sum over constant number of $m_3$ instead of $\mathcal{O}(L)$. This gives a reduction of $L$ in encoding to tensor product rep. Similarly in the decoding step, we see for fixed $m_3$ we only need to sum over $\mathcal{O}(L)$ pairs of $m_1, m_2$. This gives a reduction of $L$ as well in decoding back into irreps.

## E.4. Asymptotic runtimes in different settings

*Table 4.* Asymptotic runtimes of various tensor products for different output settings. The best performing tensor products for each output settings are highlighted in green. In the MIMO setting, the Clebsch-Gordan tensor products are highlighted in red to indicate that they can output irreps with multiplicity $> 1$, unlike the Gaunt tensor products.

| Tensor Product | SISO | SIMO | MIMO |
|---|---|---|---|
| Clebsch-Gordan (Naive) | $\mathcal{O}(L^3)$ | $\mathcal{O}(L^4)$ | $\mathcal{O}(L^6)$ |
| Clebsch-Gordan (Sparse) | $\mathcal{O}(L^2)$ | $\mathcal{O}(L^3)$ | $\mathcal{O}(L^5)$ |
| Gaunt (Original) | $\mathcal{O}(L^2 \log L)$ | $\mathcal{O}(L^3)$ | $\mathcal{O}(L^3)$ |
| Gaunt (Naive Grid) | $\mathcal{O}(L^2 \log L)$ | $\mathcal{O}(L^3)$ | $\mathcal{O}(L^3)$ |
| Gaunt (S2FFT Grid) | $\mathcal{O}(L^2 \log L)$ | $\mathcal{O}(L^2 \log^2 L)$ | $\mathcal{O}(L^2 \log^2 L)$ |
| Matrix (Naive) | $\mathcal{O}(L^3)$ | $\mathcal{O}(L^4)$ | $\mathcal{O}(L^4)$ |
| Matrix (Sparse) | $\mathcal{O}(L^3)$ | $\mathcal{O}(L^3)$ | $\mathcal{O}(L^3)$ |

# F. Expressivity

Here, we analyze the expressivity, as defined in Definition 2.2, of the various tensor products. In this case, we assume we use a tensor product to construct bilinear maps

$$B : (0 \oplus \ldots \oplus L) \times (0 \oplus \ldots \oplus L) \to (0 \oplus \ldots \oplus 2L)$$

by inserting equivariant linear layers before and after the tensor product. This choice of input and output irreps for our bilinearity is inspired by what is commonly found in practice. It is often the case that we have the same number of copies of each irrep type for our features. Since multiplicity of a rep only affects expressivity by a scaling factor, we focus on the case where there is a single copy of each irrep type up to some cutoff $L$.

By Schur's lemma, we can only linear maps between irreps of the same type and these maps must be identity. Hence, the total number of inputs and output irreps to our tensor product gives the degrees of freedom for paramterizing the linear layers from $0 \oplus \ldots \oplus L$ and to $0 \oplus \ldots \oplus 2L$. There is an additional 2-fold redundancy in overall scaling so `#Input irreps + #Ouput irreps − 2` gives an upper bound on expressivity.

## F.1. Clebsch-Gordan tensor product

In the case of CGTP, we assume input which is a single copy of each irrep up to order $L$ for $\mathcal{O}(L)$ irreps in the input. In general, tensor products of single pairs of irreps gives $\mathcal{O}(L)$ output irreps. There are $\mathcal{O}(L^2)$ pairs for a total of $\mathcal{O}(L^3)$ output irreps.

## F.2. Gaunt tensor product

In the case of GTP, we note that coefficients of spherical harmonics corresponds to single copies of each irrep. Hence, we have $\mathcal{O}(L)$ input irreps. Similarly, in the output there is only one copy of each irrep we obtain from the spherical harmonic coefficients. By selection rules, the highest order harmonic we could obtain is of order $2L$. Hence the number of output irreps is also $\mathcal{O}(L)$.

## F.3. Matrix tensor product

In the case of FTP, we encode single copies of irreps into a tensor product rep $(L/2) \otimes (L/2)$. Hence there are $\mathcal{O}(L)$ inputs. We then perform matrix multiplication which results into a $(L/2) \otimes (L/2)$ tensor product rep. But this decomposes into $0 \oplus \ldots \oplus L$ giving $\mathcal{O}(L)$ output irreps.

# G. CGTP Sparse Algorithm

While leveraging sparsity of the Clebsch-Gordan coefficients will improve asymptotic runtime, in practice we would like an implementation which is GPU friendly. Here we present an algorithm which uses the sparsity to create a constant number of generalized convolution operations.

---

**Algorithm 1** CGTP sparse

---

**Require:** Irrep 1 $\mathbf{x}^{(\ell_1)}$, Irrep 2 $\mathbf{y}^{(\ell_2)}$, Clebsch-Gordan coefficients $C^{\ell_3,m_3}_{\ell_1,m_1,\ell_2,m_2}$

  **for** $m_3 = -\ell_3,\ldots,\ell_3$ **do:**
    **for** $m_1 = -\ell_1,\ldots,\ell_1$ **do:**
      $A^{\ell_3,m_3}_{\ell_1,m_1,\ell_2} \leftarrow C^{\ell_3,m_3}_{\ell_1,m_1,\ell_2,m_1+m_3}$
      $C^{\ell_3,m_3}_{\ell_1,m_1,\ell_2,m_1+m_3} \leftarrow 0$

  **for** $m_3 = -\ell_3,\ldots,\ell_3$ **do:**
    **for** $m_1 = -\ell_1,\ldots,\ell_1$ **do:**
      $B^{\ell_3,m_3}_{\ell_1,m_1,\ell_2} \leftarrow C^{\ell_3,m_3}_{\ell_1,m_1,\ell_2,m_1-m_3}$
      $C^{\ell_3,m_3}_{\ell_1,m_1,\ell_2,m_1-m_3} \leftarrow 0$

  **for** $m_3 = -\ell_3,\ldots,\ell_3$ **do:**
    **for** $m_1 = -\ell_1,\ldots,\ell_1$ **do:**
      $C^{\ell_3,m_3}_{\ell_1,m_1,\ell_2} \leftarrow C^{\ell_3,m_3}_{\ell_1,m_1,\ell_2,-m_1+m_3}$
      $C^{\ell_3,m_3}_{\ell_1,m_1,\ell_2,-m_1+m_3} \leftarrow 0$

  **for** $m_3 = -\ell_3,\ldots,\ell_3$ **do:**
    **for** $m_1 = -\ell_1,\ldots,\ell_1$ **do:**
      $D^{\ell_3,m_3}_{\ell_1,m_1,\ell_2} \leftarrow C^{\ell_3,m_3}_{\ell_1,m_1,\ell_2,-m_1-m_3}$
      $C^{\ell_3,m_3}_{\ell_1,m_1,\ell_2,-m_1-m_3} \leftarrow 0$

  **for** $m_3 = -\ell_3,\ldots,\ell_3$ **do**
    **for** $m_1 = -\ell_1,\ldots,\ell_1$ **do**
      $\mathbf{z}^{(\ell_3)}_{m_3} \leftarrow \mathbf{z}^{(\ell_3)}_{m_3} + A^{\ell_3,m_3}_{\ell_1,m_1,\ell_2}\mathbf{x}^{(\ell_1)}_{m_1}\mathbf{y}^{(\ell_2)}_{m_1+m_3}$

  **for** $m_3 = -\ell_3,\ldots,\ell_3$ **do**
    **for** $m_1 = -\ell_1,\ldots,\ell_1$ **do**
      $\mathbf{z}^{(\ell_3)}_{m_3} \leftarrow \mathbf{z}^{(\ell_3)}_{m_3} + B^{\ell_3,m_3}_{\ell_1,m_1,\ell_2}\mathbf{x}^{(\ell_1)}_{m_1}\mathbf{y}^{(\ell_2)}_{m_1-m_3}$

  **for** $m_3 = -\ell_3,\ldots,\ell_3$ **do**
    **for** $m_1 = -\ell_1,\ldots,\ell_1$ **do**
      $\mathbf{z}^{(\ell_3)}_{m_3} \leftarrow \mathbf{z}^{(\ell_3)}_{m_3} + C^{\ell_3,m_3}_{\ell_1,m_1,\ell_2}\mathbf{x}^{(\ell_1)}_{m_1}\mathbf{y}^{(\ell_2)}_{-m_1+m_3}$

  **for** $m_3 = -\ell_3,\ldots,\ell_3$ **do**
    **for** $m_1 = -\ell_1,\ldots,\ell_1$ **do**
      $\mathbf{z}^{(\ell_3)}_{m_3} \leftarrow \mathbf{z}^{(\ell_3)}_{m_3} + D^{\ell_3,m_3}_{\ell_1,m_1,\ell_2}\mathbf{x}^{(\ell_1)}_{m_1}\mathbf{y}^{(\ell_2)}_{-m_1-m_3}$

  **return** $\mathbf{z}^{(\ell_3)}$

---

## H. Simulating the Fully-Connected Clebsch-Gordan Tensor Product with Gaunt Tensor Products

One way to increase the expressivity of GTP is to first reweight the inputs $\mathbf{x}, \mathbf{y}$. That is, we first create

$$\mathbf{x}'^{(\ell)} = a_\ell \mathbf{x}^{(\ell)} \tag{38}$$

$$\mathbf{y}'^{(\ell)} = b_\ell \mathbf{y}^{(\ell)}. \tag{39}$$

where $a_\ell$ and $b_\ell$ are learnable weights. We then perform GTP after this reweighting and extract some output irrep(s) $\ell_3$. That is we get

$$(\mathbf{x}' \otimes_{\mathrm{GTP}} \mathbf{y}')^{(\ell_3)}. \tag{40}$$

The analogous operation is fully connected CGTP. There may be multiple pairs of irreps which give a $\ell_3$ output. We can always weight and sum these to get

$$\sum_{\ell,\ell'} w_{\ell,\ell'}(\mathbf{x}^{(\ell)} \otimes_{\mathrm{CG}} \mathbf{y}^{(\ell')})^{(\ell_3)} \tag{41}$$

where $w_{\ell,\ell'}$ are learnable weights.

However, even if we only care about symmetric tensor products, the weighted GTP operation is strictly less expressive than fully connected CGTP.

More concretely, suppose we have nontrivial $\ell = 2$ and $\ell = 4$ data in our inputs. From CGTP and the selection rules we see that

$$(\mathbf{x}^{(2)} \otimes_{\mathrm{CG}} \mathbf{y}^{(2)})^{(2)} \qquad (\mathbf{x}^{(2)} \otimes_{\mathrm{CG}} \mathbf{y}^{(4)})^{(2)} \tag{42}$$

$$(\mathbf{x}^{(4)} \otimes_{\mathrm{CG}} \mathbf{y}^{(2)})^{(2)} \qquad (\mathbf{x}^{(4)} \otimes_{\mathrm{CG}} \mathbf{y}^{(4)})^{(2)} \tag{43}$$

are all nonzero. In particular, it is possible to create a $\ell = 2$ output of

$$(\mathbf{x}^{(2)} \otimes_{\mathrm{CG}} \mathbf{y}^{(2)})^{(2)} + (\mathbf{x}^{(4)} \otimes_{\mathrm{CG}} \mathbf{y}^{(4)})^{(2)}$$

with a fully connected CGTP. However, GTP instead gives a single $\ell = 2$ output of form

$$c_{2,2}^2(\mathbf{x}'^{(2)} \otimes_{\mathrm{CG}} \mathbf{y}'^{(2)})^{(2)} + c_{2,4}^2(\mathbf{x}'^{(2)} \otimes_{\mathrm{CG}} \mathbf{y}'^{(4)})^{(2)} + c_{4,2}^2(\mathbf{x}'^{(4)} \otimes_{\mathrm{CG}} \mathbf{y}'^{(2)})^{(2)} + c_{4,4}^2(\mathbf{x}'^{(4)} \otimes_{\mathrm{CG}} \mathbf{y}'^{(4)})^{(2)} \tag{44}$$

where the $c$'s are nonzero coefficients. Note that in order to have nonzero $(\mathbf{x}^{(2)} \otimes_{\mathrm{CG}} \mathbf{y}^{(2)})^{(2)}$ and $(\mathbf{x}^{(4)} \otimes_{\mathrm{CG}} \mathbf{y}^{(4)})^{(2)}$ contributions, $a_2, b_2, a_4, b_4$ must all be nonzero. However, that means we must have nonzero $(\mathbf{x}^{(2)} \otimes_{\mathrm{CG}} \mathbf{y}^{(4)})^{(2)}$ and $(\mathbf{x}^{(4)} \otimes_{\mathrm{CG}} \mathbf{y}^{(2)})^{(2)}$ contributions. Therefore weighted GTP is not expressive enough to output $(\mathbf{x}^{(2)} \otimes_{\mathrm{CG}} \mathbf{y}^{(2)})^{(2)} + (\mathbf{x}^{(4)} \otimes_{\mathrm{CG}} \mathbf{y}^{(4)})^{(2)}$, as it will necessarily mix additional terms.

## I. Details of Message-Passing Network

---
**Algorithm 2** LEARNABLETENSORPRODUCT

---
**Require:** Tensor Product $\otimes$, Number of Channels $C$ (for Gaunt tensor product).
  **procedure** LEARNABLETP($\mathbf{x}_1, \mathbf{x}_2$)
    **if** $\otimes = \otimes_{\mathrm{CG}}$ **then**
      **return** LINEAR($\mathbf{x}_1 \otimes_{\mathrm{CG}} \mathbf{x}_2$)
    **if** $\otimes = \otimes_{\mathrm{GTP}}$ **then**
      **for** $i = 1, 2, \ldots, C$ **do**
        $\mathbf{x}_1^{(i)} \leftarrow \mathrm{LINEAR}_1^{(i)}(\mathbf{x}_1)$
        $\mathbf{x}_2^{(i)} \leftarrow \mathrm{LINEAR}_2^{(i)}(\mathbf{x}_2)$
        $\mathbf{x}_o^{(i)} \leftarrow \mathrm{LINEAR}_o^{(i)}(\mathbf{x}_1^{(i)} \otimes_{\mathrm{GTP}} \mathbf{x}_2^{(i)})$
      **return** CONCATENATE($\{\mathbf{x}_o^{(i)} \mid i \in \{1, 2, \ldots, C\}\}$)
  **return** LearnableTP

---

In Algorithm 2, we create learnable (ie, parametrized) variants of the purely functional tensor products. For the Clebsch-Gordan tensor product $\otimes_{\mathrm{CG}}$, we simply add a linear layer to its output. For the Gaunt tensor product $\otimes_{\mathrm{GTP}}$, we create multiple channels, perform the tensor product channel-wise and then concatenate all irreps. This allows the output to have irreps of multiplicity $> 1$, even with the Gaunt tensor product. We set the number of channels $C$ as 4 in all experiments with the Gaunt tensor product.

In Algorithm 3, we use these learnable tensor products in a simple message-passing network, very similar to NequIP (Batzner et al., 2022).

## J. Tetris Experiment Details

The pieces are normalized such that the side length of each cube is 1. When represented as a graph, the center of each cube is a node. We instantiate the network with $d_{\max} = 1.1$ so that the centers are connected only to its immediately adjacent centers. The network finally outputs $\mathbf{x} = 7 \times 0e + 1 \times 0o$ irreps. (As a reminder, $0e$ are scalars and $0o$ are pseudoscalars).

---

**Algorithm 3** Operation of our Message Passing Neural Network

---

**Require:** Graph $G$, Message Passing Iterations $T$, Cutoff $d_{\max}$, Spherical Harmonic Degree $\ell$, Tensor Product $\otimes$

Compute neighbor lists for each node in $G$:

$$(u, v) \in E \iff \|\mathbf{r}_u - \mathbf{r}_v\| \leq d_{\max}$$

Create LEARNABLETENSORPRODUCT from $\otimes$.

**for** $v \in V$ **do**:
$\quad h_v^{(0)} \leftarrow [1]$

**for** $t = 1, 2, \ldots, T$ **do**:
$\quad$ **for** $v \in V$ **do**:
$\quad\quad h_v^{(t)} \leftarrow \frac{1}{|\mathcal{N}(v)|} \sum_{u \in \mathcal{N}(v)} \text{MLP}(\|\mathbf{r}_u - \mathbf{r}_v\|) \times \text{LEARNABLETENSORPRODUCT}(h_u^{(t-1)}, Y_\ell(\mathbf{r}_u - \mathbf{r}_v))$
$\quad\quad h_v^{(t)} \leftarrow \text{GATE}(h_v^{(t)})$
$\quad\quad h_v^{(t)} \leftarrow \text{CONCATENATE}([h_v^{(t-1)}, h_v^{(t)}])$
$\quad\quad h_v^{(t)} \leftarrow \text{LINEAR}(h_v^{(t)})$

**return** $\{h_v^{(T)}\}_{v \in V}$

---

The logits and predicted probabilities are then computed by:

$$l_0 = \mathbf{x}^{(0o)} \times \mathbf{x}^{(0e)_0} \tag{45}$$

$$l_1 = -\mathbf{x}^{(0o)} \times \mathbf{x}^{(0e)_0} \tag{46}$$

$$l_i = \mathbf{x}^{(0e)_i} \quad \text{for } i \geq 2 \tag{47}$$

$$p_i = \text{softmax}(l_i) \tag{48}$$

It is clear that defining the logits in this manner preserves the rotational and reflection symmetries. The predictions are clearly invariant under rotations (as they are $\ell = 0$ irreps), and under reflections: $\mathbf{x}^{(0o)} \to -\mathbf{x}^{(0o)}$ but $\mathbf{x}^{(0e)_i} \to \mathbf{x}^{(0e)_i}$.

We set the number of message-passing steps $T$ to be 3, to allow the interactions $1o \otimes 1o \to 1e$ and then $1e \otimes 1o \to 0o$, so the pseudoscalar can be created. The degree of spherical harmonics is kept as $\ell = 4$. The irreps of the hidden layers are restricted to some cutoff $L$, which is varied from 1 to 4 to vary the expressivity of the network. We train the model with the Adam optimizer with learning rate 0.01 to minimize the standard cross-entropy loss to one-hot encoded labels for the 8 pieces.

## K. Proof of Theorem 3.2

*Proof.* It is known (Gaunt, 1929) that:

$$Y_{\ell_1}^{m_1} \cdot Y_{\ell_2}^{m_2} \propto \sum_{j,\ell} C_{\ell_1,0,\ell_2,0}^{\ell,0} C_{j_1,m_1,j_2,m_2}^{j,m} Y_\ell^m.$$

Hence, we see that

$$(\mathbf{x}^{(\ell_1)} \otimes_{\text{GTP}} \mathbf{y}^{(\ell_2)})^{(\ell)} \propto C_{\ell_1,0,\ell_2,0}^{\ell,0} C_{j_1,m_1,j_2,m_2}^{j,m} (\mathbf{x}^{(\ell_1)} \otimes_{\text{CG}} \mathbf{y}^{(\ell_2)})^{(\ell)}.$$

For selection rule 1, this is inherited from the selection rules of CGTP. For selection rule 2, this follows from the selection rules for the $C_{\ell_1,0,\ell_2,0}^{\ell,0}$ term which is nonzero only when $\ell_1 + \ell_2 + \ell$ is even. $\qquad\square$

## L. Details of Benchmarking Setup

### L.1. GPU

**Wall-Clock Time:** Total wall-clock time is reported as the sum of `gpu_time_duration.sum` metric for all the kernels executed for that function call. We confirmed that this closely matches the `jax` wall-clock time with some profiling overhead due to kernel replays.

**FLOPs:** We obtain the list of all FLOPS generating instructions that are executed within each kernel run on the GPU. We then count all instructions and multiply them by their specific weighing factors (the number of FLOPs in each instruction). The weighing factor for a multiplication or addition is 1 and the weight for a multiply-and-add (FMA) is 2. For Tensor Core instructions on an Ampere GPU, the peak performance is 1024 FMA/cycle/SM and since 1 FMA = 2 FLOPS, we get 2048 (TensorCoreWeight, 2023). We check the consistency of this weighing factor by running a matrix-multiplication benchmark. Further details can be found in Empirical Roofline Toolkit (Yang et al., 2020).

**Average Throughput (FLOPs/s):** We first calculate the Tensor Cores and CUDA Cores Throughput for every kernel by dividing the FLOPs for that kernel by the time taken to execute that kernel measured using `gpu_time_duration.sum`. We report the average throughput for a range of kernels launched for a given tensor product function call.

**Average DRAM Bandwidth (GB/s):** For every kernel, we take the sum of `dram_bytes_read.sum` and `dram_bytes_write.sum` and divide it by the kernel execution time. We report the average bandwidth for a range of kernels launched for a given tensor product function call

**GPU:** We gathered the GPU plots on an NVIDIA RTX A5500 with default JAX precision (F32), running the CUDA driver version 550.90.07 and CUDA toolkit version 12.5. We used Nsight Compute 2024.2.0.0 build 34181891 and JAX version 0.4.30. JAX automatically switches to TF32 wherever appropriate.

| Precision | Metrics | Weight Factor |
|---|---|---|
| FP64 | `sm__sass_thread_inst_executed_op_dadd_pred_on.sum` | 1 |
| | `sm__sass_thread_inst_executed_op_dmul_pred_on.sum` | 1 |
| | `sm__sass_thread_inst_executed_op_dfma_pred_on.sum` | 2 |
| FP32 | `sm__sass_thread_inst_executed_op_fadd_pred_on.sum` | 1 |
| | `sm__sass_thread_inst_executed_op_fmul_pred_on.sum` | 1 |
| | `sm__sass_thread_inst_executed_op_ffma_pred_on.sum` | 2 |
| FP16 | `sm__sass_thread_inst_executed_op_hadd_pred_on.sum` | 1 |
| | `sm__sass_thread_inst_executed_op_hmul_pred_on.sum` | 1 |
| | `sm__sass_thread_inst_executed_op_hfma_pred_on.sum` | 2 |
| Tensor Core | `sm__inst_executed_pipe_tensor.sum` | 2048 |

*Table 5.* FLOPS weights for various Nsight Compute metrics.

## L.2. CPU

**Wall-Clock Time:** The elapsed time after compiling using `jax.jit`. To enable accurate measurements, we calculate the mean wall-clock time for 100 rounds after performing 10 warmup rounds.

**FLOPs:** We use Linux's `perf` profiler to directly count the number of FLOPs through `fp_ret_sse_avx_ops.all` metric (specific to AMD).

We had to skip throughput and bandwidth because we could not colleft hardware-counters for them using `perf`

**CPU:** The CPU plots were gathered on an AMD EPYC 7313 16-core 3.7 GHz processor with default JAX precision.

# M. Additional Benchmarks

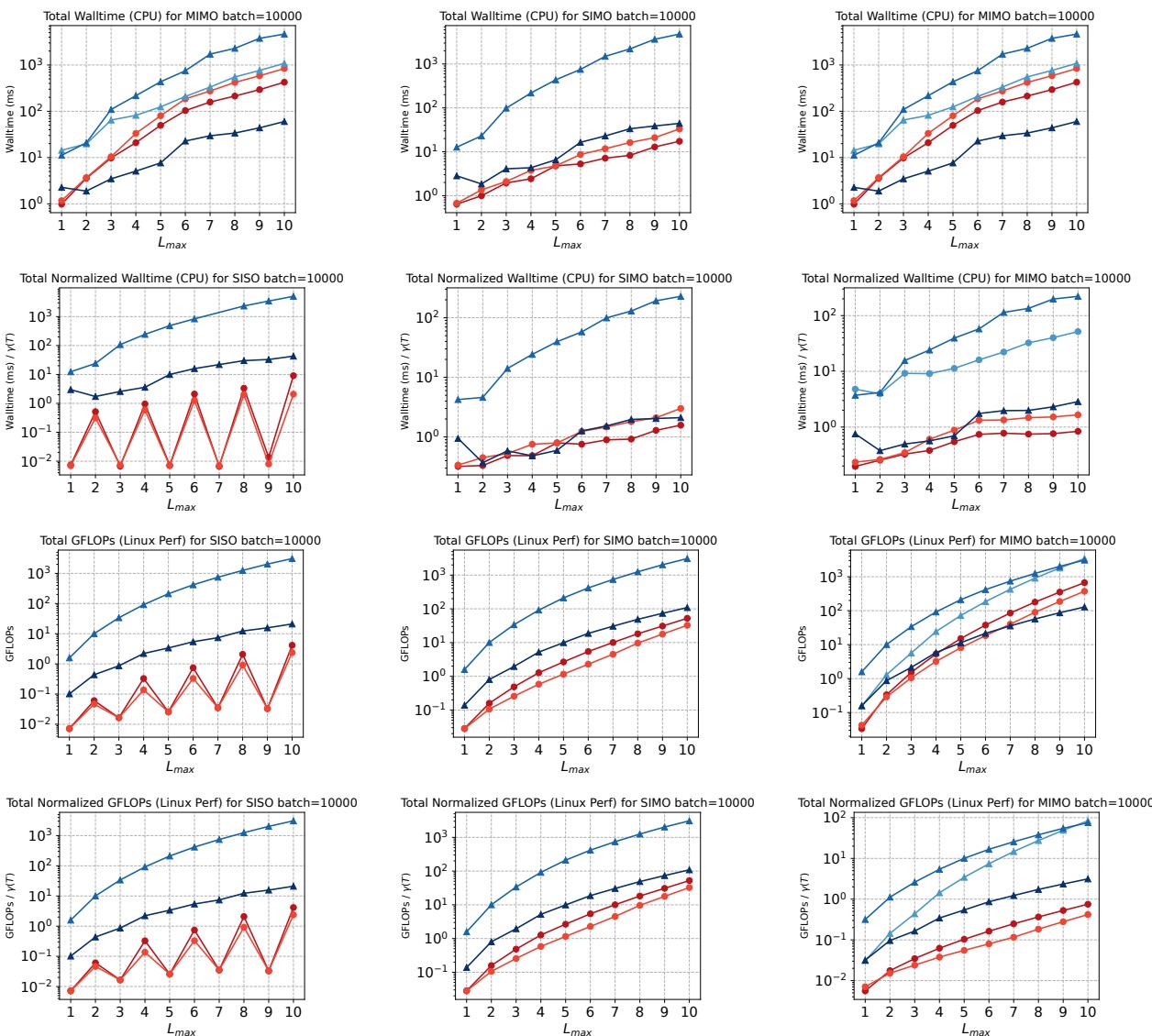

*Figure 7.* Analysis of SISO, SIMO and MIMO (Table 4 for input settings) performance for different tensor products on AMD EPYC 7313 : Total walltime (top row), Total normalized walltime (second row), Total GFLOPs (third row) and Total normalized GFLOPs (bottom row). Note that MTP only supports MIMO. We had to skip values due to profiling errors.

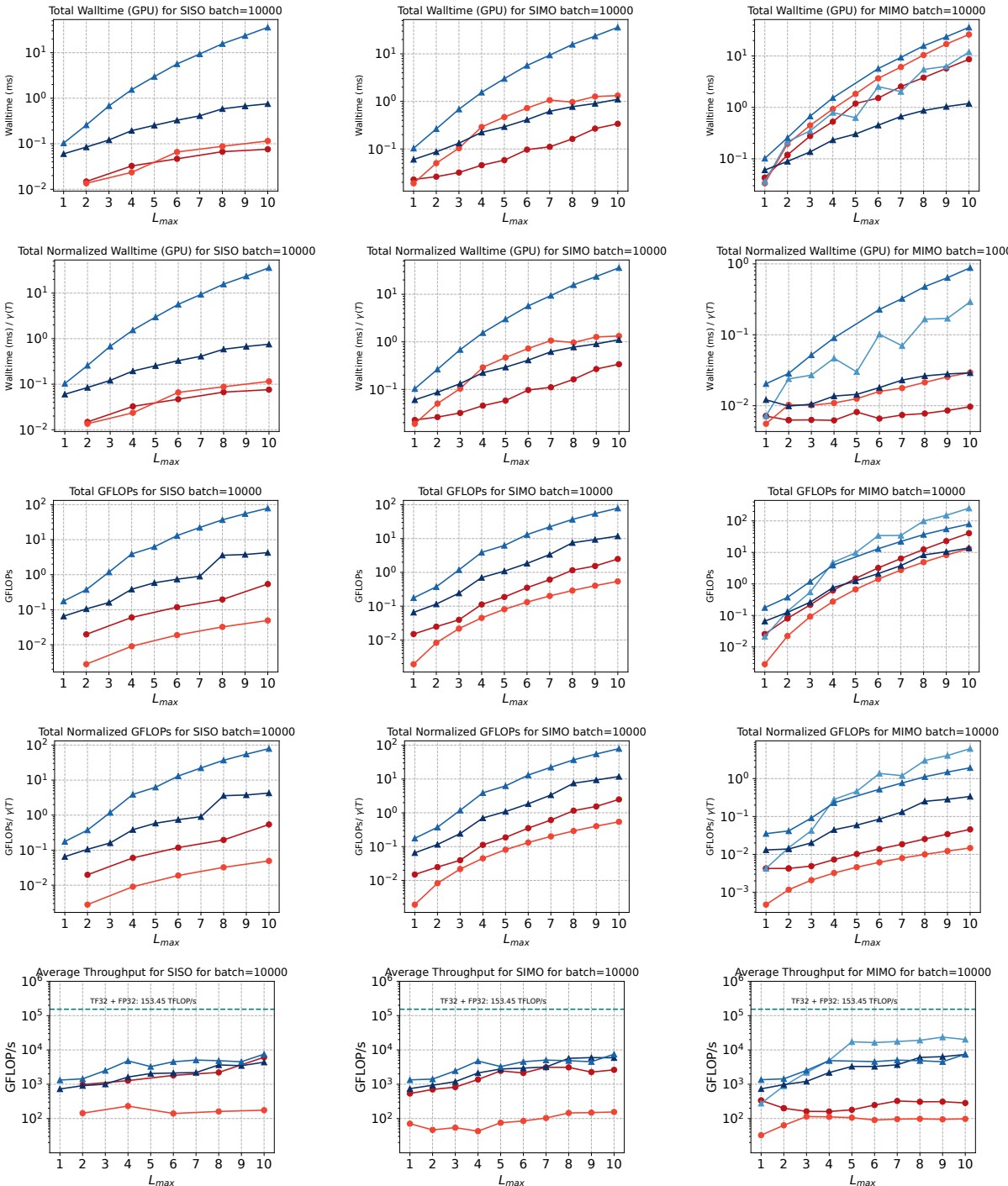

*Figure 8.* Analysis of SISO, SIMO and MIMO (Table 4 for input settings) performance for different tensor products on RTX A5500 : Total forward walltime (top row), Total forward normalized walltime (second row), Total GFLOPs (third row), Total forward normalized GFLOPs (fourth row) and Average forward throughput (bottom row). Note that MTP only supports MIMO. We had to skip values due to profiling errors.

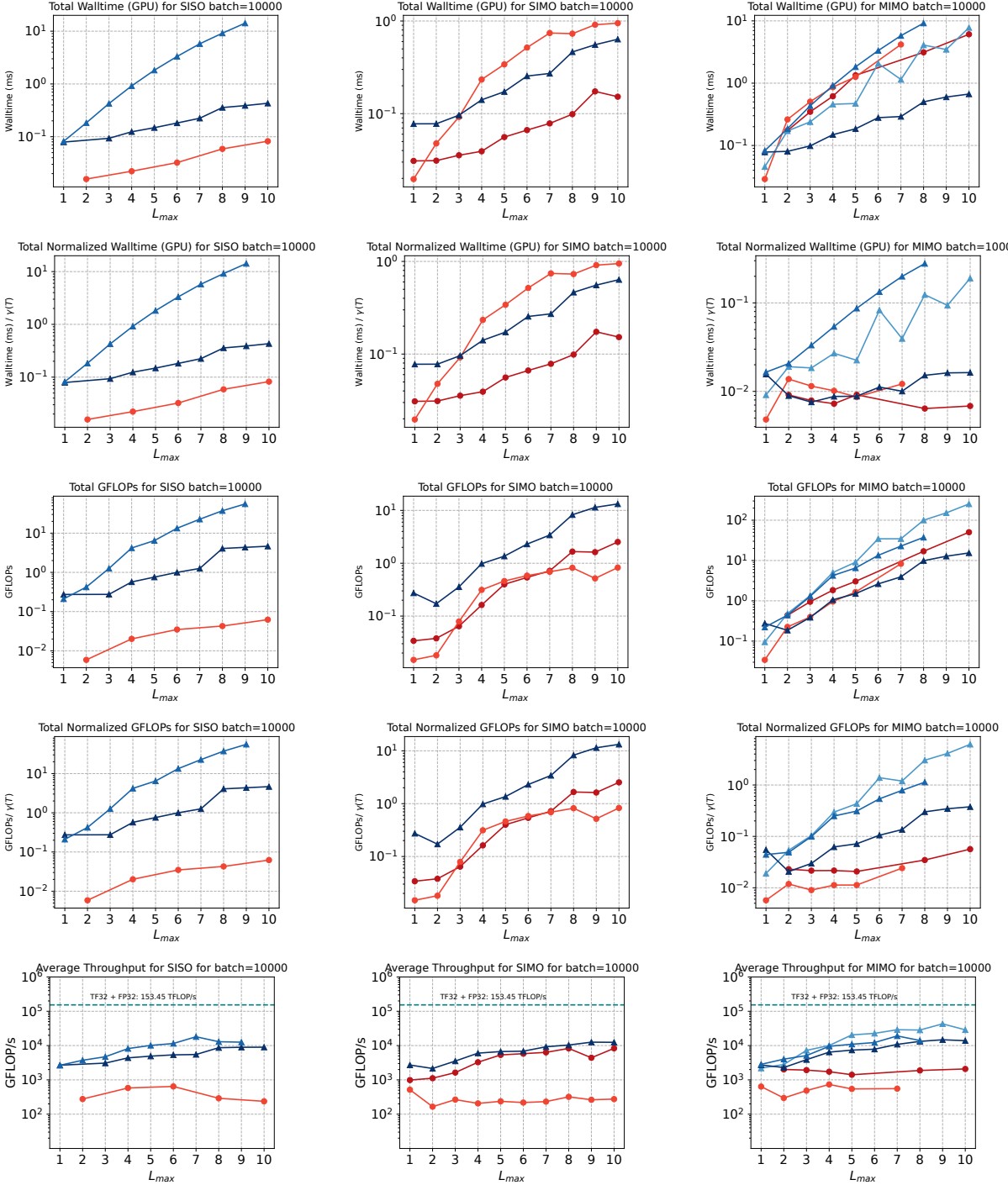

*Figure 9.* Analysis of SISO, SIMO and MIMO (Table 4 for input settings) performance for different tensor products on A100 GPU, showing Total forward walltime (top row), Total normalized forward walltime (second row), Total forward GFLOPs (third row) and Total forward normalized GFLOPs (fourth row) and Average forward throughput (bottom row). Note that MTP only supports MIMO. We had to skip values due to profiling errors.

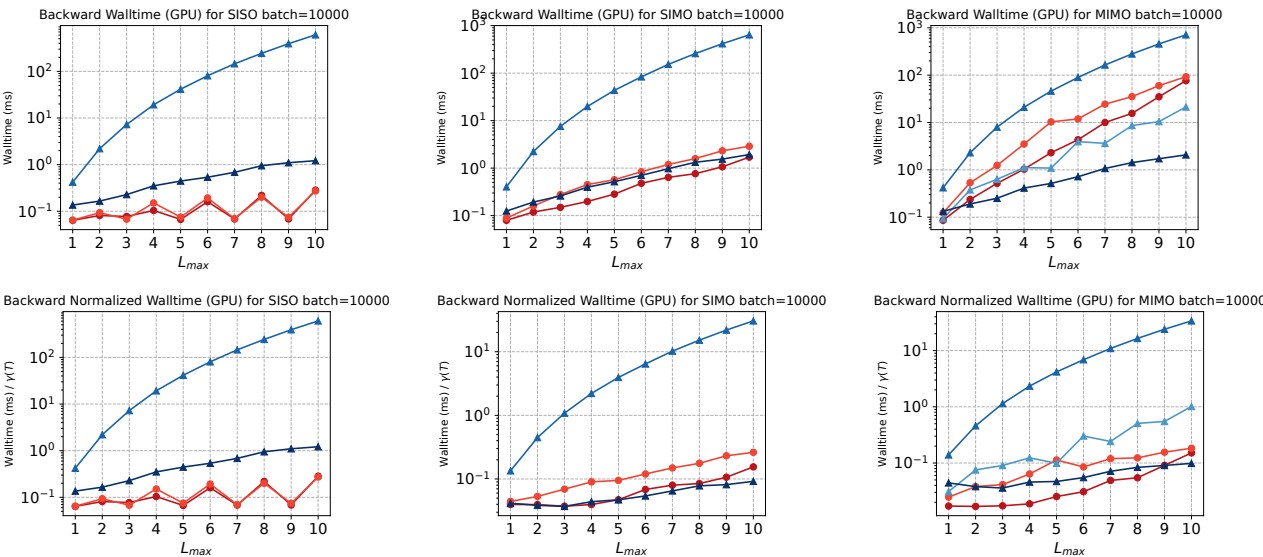

*Figure 10.* Analysis of SISO, SIMO and MIMO (Table 4 for input settings) performance for different tensor products on RTX A5500 : Total backward walltime (top row), Total backward normalized walltime (bottom row). Note that MTP only supports MIMO

