# OpenReview forum: "The Price of Freedom: Exploring Expressivity and Runtime Tradeoffs in Equivariant Tensor Products"
_ICML.cc/2025/Conference — ICML 2025 poster_

### Official Review · Reviewer_Hd64 · 2025-02-20

**Overall Recommendation:** 2

**Summary:**

This paper analyzes various tensor product operations in equivariant neural networks for 3D modeling. It introduces measures of expressivity and interactability and improves the Gaunt tensor product (GTP) with a spherical grid, achieving a 30% speedup. The paper also presents microbenchmarks, showing discrepancies between theoretical and empirical runtime, emphasizing the need for application-specific benchmarking.

**Claims And Evidence:**

The content has already been provided in the subsequent subsections.

**Essential References Not Discussed:**

No.

**Experimental Designs Or Analyses:**

The content has already been provided in the subsequent subsections.

**Methods And Evaluation Criteria:**

The content has already been provided in the subsequent subsections.

**Other Comments Or Suggestions:**

N/A.

**Other Strengths And Weaknesses:**

Strengths:

1. The paper provides an extensive theoretical introduction and comparison of different types of tensor products, making it easy for readers to understand the differences between them.
2. The paper introduces a new evaluation method for tensor products, enhancing our understanding of their relative efficiency and expressivity, which can help in selecting the appropriate type of tensor product for networks.

Weaknesses:

1. The problem the paper aims to solve is not entirely clear. Specifically, while a new evaluation criterion for tensor-product computations is introduced, the paper does not explain what additional benefits this provides for tensor-product-based equivariant models. For example, is the model's runtime/accuracy proportional to the tensor-product's runtime/expressivity as proposed? Can tensor product types be evaluated solely based on this criterion without model training? The paper also lacks experiments to verify this point.

2. Considering the theoretical improvement made to the Gaunt Tensor Product, the 30% speedup of S2grid should be seen as marginal. Compared to other acceleration works like Gaunt Tensor Product and cuEquivariance, this improvement seems relatively modest.

3. Regarding the evaluation of different tensor products, the actual performance of these tensor products also depends on the degree of optimization applied, which greatly impacts the scaling constants. It is necessary to discuss or clarify the implementation method in the paper. Otherwise, such comparisons would be unfair and could not be strictly aligned with the theoretical complexity.

**Questions For Authors:**

1. Based on Section 6.2, the lack of antisymmetry in GTP seems to significantly affect its performance. Should further experiments be conducted on general datasets to explore this issue? Additionally, how do the evaluation criteria proposed in the paper inspire solutions for addressing this problem?

2. For single tensor products not embedded in networks, performance benchmarking often faces challenges such as very short execution times or inaccurate timing tools. How did you address these issues in your actual experiments?

3. Overall, the paper gives the impression that it touches on many subtopics but provides only a superficial investigation of each, including the evaluation criteria, improvements to GTP, and the study of antisymmetry. Have you considered organizing these topics into separate papers, each with more detailed experiments, such as embedding the tensor products into networks and testing results on popular small-molecule or materials datasets?

**Relation To Broader Scientific Literature:**

No.

**Theoretical Claims:**

The content has already been provided in the subsequent subsections.

---

> ### Author Rebuttal · Authors · 2025-04-01
>
> We thank the reviewer for your reading of our work and feedback. We appreciate that the reader finds our explanation of differences between TPOs clear.
>
> Regarding the weaknesses:
> ## Weaknesses
> 1. We would like to clarify that the main point of the paper is understanding how alternatives to CGTP are achieving their performance gains. Crucially, we want to clarify that direct speed comparisons between GTP and CGTP are unfair. Instead, **GTP is performing a different operation from CGTP**. This realization was a major motivation for our work and in fact through our work we see that GTP does not give speed ups compared to CGTP in a fairer comparison.
>
>      The key point is our framework provides a **systematic way** to analyze the differences between any TPOs (such as GTP, MTP, CGTP) and compare speedups in a fairer setting. We believe that this is extremely valuable for designing new TPOs.
>
>      We do not claim that our measure of expressivity has direct correlation with actual training performance. In fact, combining the previous observation that GTP can achieve similar performance as CGTP and our result showing it **provably** has fewer degrees of freedom is already interesting. The **true benefit of GTP should not be understood as a speedup but rather the elimination of many degrees of freedom with minimal impact**. These observations illuminate potential avenues for further improvement.
>
> 2. By identifying that we can directly use a S2Grid, we highlight a possible asymptotic runtime improvement over the original implementation of GTP by using a S2FFT. This is the **first** algorithmic improvement which cannot simply be explained by a reduction in degrees of freedom. We believe this is quite interesting and can potentially have a significant impact in the future.
>
>      However, we emphasize that the current $\ell$s used are far too small to see these asymptotic benefits from the complicated S2FFT algorithm. Despite this, using a S2Grid and a seminaive S2 Fourier transform (instead of S2FFT) gives a much simpler implementation of GTP which already sees performance gains over the original.
>
> 3. This is a good point, and we did try to mitigate implementation differences as much as possible (e.g. using JAX for everything). We would like to highlight that it is not the results themselves but the method of benchmarking that is important.
>
>      Our microbenchmarks provide valuable insight on how different implementations can be improved. In particular, the discrepancy between FLOP counts (which are invariant) and walltime indicates potential for significant acceleration (eg. custom kernels) by better utilizing the GPU.
>
> ## Questions
> 1. This is a great question! Our experiment in 6.2 gives a simple demonstration highlighting this failure. However, we actually strongly suspect most common molecular datasets are minimally impacted. Loosely speaking, this is because irrep types of commonly predicted quantities such as energy or forces can be constructed purely from symmetric tensor products of the input irrep types. We are actively exploring the impact of antisymmetric tensor products in future work.
>
>      Further, we have come up with a new GTP-like tensor product which does not suffer the antisymmetry issues and has the same potential asymptotic benefit from S2FFT. However, we feel it distracts from the main message of this paper and have omitted it.
>
> 2. Thanks for the question! We chose the highest batch size (10,000) that we could fit onto the GPU without going out of memory. To offset any profiling overheads we chose to measure our walltime/FLOPs/throughput metrics using hardware instruction counters at the GPU driver level which is based on Roofline Toolkit Framework for Deep Learning (https://arxiv.org/abs/2009.05257) and has been successfully used in FourCastNet (https://dl.acm.org/doi/10.1145/3592979.3593412).
>
>
> 3. We are providing a framework for systematically analyzing how different TPOs achieve efficiency gains. Theoretically, we provide a way to see whether these came from a clever reduction in degrees of freedom or from actual algorithmic improvements. Our microbenchmarks provide a way to see runtimes of TPOs in practice and avenues for implementation improvements.
>
>      Many previous works simply state their method is faster and achieves similar training results. Our work provides a way to analyze why their method was faster and whether there may be possible improvements or limitations. Our experiments are designed with this goal in mind. As such, the comparison of different methods in training results is not the central focus of this work (those results can be found in the original papers introducing the specific TPOs). However we understand the importance of training performance and agree that evaluation of training performance is also crucial for new TPO proposals.

---

> > ### Comment · Reviewer_Hd64 · 2025-04-08
> >
> > I sincerely appreciate the authors' detailed response, especially the clarification and explanation regarding the theme of the paper. Overall, I fully recognize the theoretical contribution of this work in the analysis of tensor product operations, and I will raise my socre to 2 while my main concerns remain.
> >
> > Firstly, similar to the views expressed by Reviewers a8uP and t24S, the novelty of the proposed GTP and its performance gains appear to be limited. If the authors attribute this to the current maximal order being too small, then the question arises: would higher orders be truly beneficial for practical applications? According to some literature like equiforme v2, increasing the maximal order further seems to yield diminishing returns, which poses limitations for the applications.
> >
> > In addition, regarding the benchmarking methodology and the reported results:
> > First, FLOP counts are not invariant. Given that certain computations can be reused (as is indeed the case in mainstream libraries such as e3nn and cuEquivariance), this can significantly affect the constant factors in computation cost and, consequently, the actual runtime.
> >
> > Moreover, based on your statement that "microbenchmarks provide valuable insight on how different implementations can be improved," it would be more appropriate to perform benchmarks using official implementations from current mainstream software, or at least provide a comparison against your own simplified implementations. This helps ensure that the insights remain applicable and timely for currently used implementations. Cause engineering optimizations that reduce redundancy have been extensively studied, and even well-optimized methods can differ by orders of magnitude depending on implementation. Therefore, discussing benchmark insights in isolation from these practical factors seems of limited value.
> >
> > [1]https://developer.nvidia.com/blog/accelerate-drug-and-material-discovery-with-new-math-library-nvidia-cuequivariance
> >
> > [2]Geiger M, Smidt T. e3nn: Euclidean neural networks[J]. arXiv preprint arXiv:2207.09453, 2022.
> >
> > [3]Liao Y L, Wood B, Das A, et al. Equiformerv2: Improved equivariant transformer for scaling to higher-degree representations[J]. arXiv preprint arXiv:2306.12059, 2023.
> >
> > ——————————————————————————————————
> >
> > I appreciate the author’s further clarifications on their Rebuttal Comment Reply, and I will put my further comments here.
> >
> > I fully agree with the statement that *“higher degrees are consistently helpful”*. However, the precision gains tend to exhibit diminishing returns when weighed against the loss in computational efficiency. Therefore, most molecular models opt to cap the maximum degree at 2. For instance, the spherical grid approach benchmarks presented by the author also select MACE, which uses a maximum degree of 2. Thus, if strong performance is only observed in higher-degree networks, it may indeed limit the method’s applicability. Furthermore, I appreciate that the author provided many examples of high-order tensor methods. If the goal is to highlight the spherical grid approach’s capabilities at higher degrees, perhaps considering those methods — rather than MACE — as benchmark baselines would be more appropriate.
> >
> > Regarding the implementation details, I appreciate the additional clarifications. However, in the absence of supplementary material, I am unable to offer further comments on the technical specifics. One reason I suggested “using official implementations from current mainstream software” is that, according to the open-source official implementation of GauntTP (https://github.com/lsj2408/Gaunt-Tensor-Product), its relative wall-time compared to the CGTP implementation in `e3nn` appears to be the exact opposite of what is shown for CGTP in your Figure 4. This raises a reasonable concern that there may still be unresolved issues requiring further investigation.

---

> > > ### Author Response · Authors · 2025-04-08
> > >
> > > We thank the reviewer for reconsidering their score, and address their final questions here.
> > >
> > > For the question “would higher orders be truly beneficial for practical applications”, we provide some evidence from prior work:
> > > - the neural atomic potentials MACE (Fig. 2 in https://arxiv.org/pdf/2206.07697), NequIP (Table 2 in https://www.nature.com/articles/s41467-022-29939-5 and Fig. 6 in https://www.nature.com/articles/s41524-024-01264-z) and FLARE (Fig. 7 in https://www.nature.com/articles/s41524-024-01264-z),
> > > - charge density prediction (Fig. 2 in https://arxiv.org/pdf/2210.04766, Fig. 2 in https://iopscience.iop.org/article/10.1088/2632-2153/acb314#mlstacb314f2 and Fig. 3 in https://www.nature.com/articles/s41524-024-01343-1 from ChargE3Net),
> > > - and the autoregressive molecular generative model Symphony (Fig. 12 in https://openreview.net/pdf?id=MIEnYtlGyv#page=23.62)
> > >
> > > where we see significant benefit from increasing the maximum degree L of the irreps in the hidden layers. Indeed, even in the EquiformerV2 paper (Table 1c in https://arxiv.org/pdf/2306.12059#page=5.94), the results start from L = 4 which is already much higher than usual applications of equivariant networks. Note that the paper also makes the claim that ‘higher degrees are consistently helpful’ in the caption of Table 1c.
> > >
> > > For our implementations, we used primitives from e3nn-jax as much as possible. Note that these primitives are already significantly faster than those in e3nn-torch, even with the new torch.compile functionality. We will add a comparison between these primitives in the camera-ready version of our submission, if accepted. We also apologize for not uploading our code as supplementary information, which we will add in our camera ready version, if accepted.
> > >
> > > Importantly, we precompute all constants (such as the Clebsch-Gordan coefficients and change-of-basis matrix for the Fourier transform on S2) during compile-time. These computations are not measured in the FLOPs we report. In fact, we were very careful to choose hardware-level counters instead of relying on the FLOP counter in JAX because of a bug in XLA that gives incorrect FLOP counts for certain operations (https://github.com/openxla/xla/issues/10479).
> > >
> > > Regarding cuEquivariance, the blog linked by the reviewer shows speedups for a specific tensor product used in MACE and NequIP (as part of DiffDock). Note that cuEquivariance provides kernels for a weighted version of the tensor products we use here, which can potentially have different runtimes because of compiler optimizations. In fact, the unweighted version of their tensor product kernel (https://github.com/NVIDIA/cuEquivariance/blob/fd8484b9ae93a6866e358a16dfb3a2e5474b0524/cuequivariance/cuequivariance/group_theory/descriptors/irreps_tp.py#L86-L146) matches the operations we benchmark but performs significantly worse. We will add a discussion of this issue to the camera-ready version of our submission, if accepted.
> > >
> > > The point of our work is to create a level playing field across tensor products to appropriately normalize their input and output spaces. In particular, we wanted to decouple algorithmic improvements from engineering improvements (such as those done by cuEquivariance and OpenEquivariance).
> > >
> > > We hope this response clarifies the questions brought up by the reviewer.

---

### Official Review · Reviewer_W5ML · 2025-03-09

**Overall Recommendation:** 4

**Summary:**

The paper presents a comprehensive analysis of tensor products and tensor product operations used in $E(3)$-equivariant models based on spherical tensors, including the Clebsch-Gordan tensor product (CGTP), Gaunt tensor product (GTP), and Matrix tensor product (MTP). The authors introduce expressivity and interactability as key measures to characterize these operations, demonstrating that speedups presented in prior work often come at the cost of expressivity. Furthermore, the paper proposes a more efficient implementation of GTP using a spherical grid, demonstrating practical performance improvements.

## update after rebuttal

The authors addressed all my questions. However, I maintained my score at 4 as I remain uncertain about the broader impact of the work. Overall, I recommend this work for publication.

**Claims And Evidence:**

The paper's main claim is that tensor product operations proposed as an alternative to CGTP achieve computational efficiency at the cost of expressivity. This claim is supported through theoretical analysis and numerical experiments. The authors comprehensively compare asymptotic computational complexities and runtimes, showing that CGTP retains the highest expressivity but can be more computationally expensive. Furthermore, the paper provides empirical evidence that GTP cannot represent antisymmetric interactions using a 3D Tetris classification task.

**Essential References Not Discussed:**

The paper does not reference works discussing tensor products in bases other than the spherical one (e.g., https://arxiv.org/abs/2306.06482, https://arxiv.org/abs/2405.14253, https://arxiv.org/abs/2412.18263). Aside from this, the paper sufficiently covers related literature.

**Experimental Designs Or Analyses:**

The experimental design is sound, with well-chosen benchmarks and clear comparisons across tensor product operations. However, the benchmarks focus primarily on the forward pass, while the section title for the 3BPA data set states that atomic forces and energies were evaluated. Given the importance of gradients in applications such as machine-learned force fields, it would be beneficial to analyze backward pass performance if possible or clarify whether this analysis has already been conducted for the 3BPA data set.

**Methods And Evaluation Criteria:**

The proposed methods, such as the spherical grid implementation of GTP, and evaluation criteria, including 3BPA and 3D Tetris data sets, are well-aligned with the problem. The authors carefully assess theoretical expressivity and runtime complexity as key metrics, including rigorous numerical experiments.

**Other Comments Or Suggestions:**

1. The authors could clarify why the discussion is limited to the spherical basis and whether their framework extends to other bases.

2. It would be helpful to discuss potential extensions of the framework to “backward pass” computations.

3. Briefly mentioning how alternative bases might impact expressivity and runtime would offer a more comprehensive perspective.

Minor comments:

4. The caption of Figure 5 appears redundant, as it is a part of Figure 4.
5. "($\mathbf{x}^{(l_1)}$" appears to be missing in Equation 19.

**Other Strengths And Weaknesses:**

Please refer to the comments in previous sections for all strengths and weaknesses of the presented work.

**Questions For Authors:**

My questions would relate to the issues or comments raised above, so addressing them would suffice to change my evaluation of the paper.

**Relation To Broader Scientific Literature:**

The paper fits well with the broader literature on equivariant neural networks and tensor product operations. It thoroughly discusses prior work on Clebsch-Gordan coefficients, corresponding tensor products, and equivariant architectures based on spherical tensors. However, it does not address tensor products in alternative bases, e.g., the Cartesian basis. While the spherical basis is currently the dominant choice in the community, emerging studies on Cartesian representations could provide additional context for the paper’s claims.

**Theoretical Claims:**

The theoretical claims regarding the expressivity of different tensor product operations are well-supported. The derivations of runtime complexities and selection rules for each tensor product align with established results in the literature. The discussion on expressivity offers a clear explanation for why certain tensor products fail to capture specific interactions.

However, the statement that CGTP is the only true tensor product is arguable, especially given that the study focuses exclusively on spherical tensors. For example, an alternative tensor product can be defined in the Cartesian basis, an area with a smaller but gradually growing body of literature that deserves mention in this context; see, e.g., https://arxiv.org/abs/2306.06482, https://arxiv.org/abs/2405.14253, https://arxiv.org/abs/2412.18263, and references therein.

---

> ### Author Rebuttal · Authors · 2025-04-01
>
> We thank the reviewer for your careful reading of our work and positive feedback. We appreciate that the reviewer finds our evaluation criteria and experiments well aligned with the problem, the theoretical claims well supported, and discussion on expressivity a clear explanation for why certain TPOs fail to capture specific interactions.
>
> Regarding the comments and suggestions
> 1. We thank the reviewer for bringing our attention to recent work in the Cartesian basis. This is a very interesting body of work which we will definitely mention in our revised manuscript.
>
>      We would like to point out that our focus is on irrep based frameworks as in that case maximally expressive linear layers are easy to parameterize (as briefly explained in section 2.1). For the case of Cartesian tensors, it is easy to perform tensor products and remain a Cartesian tensor. However, especially at higher rank, it becomes difficult to parameterize fully expressive equivariant linear layers. Regarding the specific sources:
>
>      It seems the interaction in https://arxiv.org/abs/2306.06482 is limited to rank 2 tensors and is a special case of MTP. https://arxiv.org/abs/2412.18263 seems to have significant work on decomposing Cartesian tensors to irreps in order to construct equivariant linear layers.
>
>      It seems https://arxiv.org/abs/2405.14253 introduces a Cartesian tensor product which is also a true tensor product. However, this method has poor asymptotic scaling for 2-fold tensor products which was the focus of this work. Instead, their speedups are for small $\ell$ or large $\nu$-fold tensor products (performing multiple tensor products in succession like in MACE). We believe our framework can also be adapted for analyzing large $\nu$ by replacing the fixed bilinearities with fixed $\nu$-linearities and defining expressivity as the dimension of constructible $\nu$-linearites by inserting equivariant linearities in the inputs and output.
>
>      We will add a discussion of these points in our final manuscript.
>
> 2. This is a good question. We would expect the same optimizations that were accelerating the forward pass to work for the backward pass. We were able to confirm this hypothesis by running a short experiment on the different input/output irreps settings and are able to see similar trends as the forward time. The plots and experiment details are included below. In particular, we create a random $z$ and then benchmark how long it takes to compute $\nabla_x(T(x,y)-z)^2$.
>
> https://anonymous.4open.science/r/PriceofFreedom-A835/benchmarking/plots/png/walltime_bwd_gpu_MIMO_10000_RTX-1.png
> https://anonymous.4open.science/r/PriceofFreedom-A835/benchmarking/plots/png/walltime_bwd_normalized_gpu_MIMO_10000_RTX-1.png
> https://anonymous.4open.science/r/PriceofFreedom-A835/benchmarking/plots/pof_legend.png
>
> 3. We will definitely mention other basis choices in our revised manuscript and discuss Cartesian tensors in particular. The content in Section 2.1 motivates why we focus on an irrep basis.

---

> > ### Comment · Reviewer_W5ML · 2025-04-02
> >
> > I appreciate the authors' response but will keep my score at 4, as I am unsure of the work's broad impact for an oral presentation.

---

> > > ### Author Response · Authors · 2025-04-08
> > >
> > > We would like to thank you again for your time spent reviewing our work. Your feedback was very useful in helping improve our work. In addition, we will add our code as supplementary material in the camera-ready version if accepted.

---

### Official Review · Reviewer_t24S · 2025-03-10

**Overall Recommendation:** 3

**Summary:**

This paper aims to advance the fundamental understanding of equivariant neural networks by studying different mappings from the product of vector spaces into tensor product spaces (including the well-known CG tensor product), which serve as the building blocks of expressive equivariant architectures. In particular, it introduces a straightforward but novel measure to compare the expressivity of different mapping designs. Additionally, the paper discusses implementation details and provides a comprehensive analysis of each operation in terms of FLOPs, wall-clock time, GPU utilization, and expressivity.

**Claims And Evidence:**

The experiments demonstrate the runtime and expressivity, aligning well with the theoretical analysis.

**Essential References Not Discussed:**

The novelty of the new GTP implementation seems weak. First, the discrete spherical harmonic transform has already been extensively studied. In particular, this transform has a well-known convolution theorem, which appears to be closely related to the proposed implementation. However, it is unclear whether the authors have adequately addressed the overlap with existing works or provided relevant references from related fields.

[1] Blais, J. R. (2008, June). Discrete spherical harmonic transforms: Numerical preconditioning and optimization. In International Conference on Computational Science (pp. 638-645). Berlin, Heidelberg: Springer Berlin Heidelberg.

**Experimental Designs Or Analyses:**

The experimental design is sound.

**Methods And Evaluation Criteria:**

The proposed measure of expressivity for different mappings into tensor product spaces seems standard and reasonable. The runtime evaluation is rigorous and comprehensive, whereas the assessment of expressivity appears relatively adequate.

**Other Comments Or Suggestions:**

I believe I understand the definition of the selection rule, where $c_*$ appears to represent the multiplicity of irreducible representations in different $G$-spaces. However, in Proposition 3.1, $c_Z$ is a tuple rather than a single number. It seems that the goal is to separate the interaction outputs—essentially, different tuples correspond to different indices. If my interpretation is correct, I believe the definition and related explanations could be revised to improve clarity for a broader audience.

**Other Strengths And Weaknesses:**

Weaknesses:
- The contribution of the new GTP implementation appears limited (see my comment above).
- The evaluation of expressivity is constrained by the simplicity and small size of the dataset used. It would be highly beneficial to see a discussion of the trade-offs when applied to a large-scale real-world dataset. I believe the implications and potential impact of this work are limited by its current application to a relatively simple dataset.

**Questions For Authors:**

The theoretical results and discussion on runtime appear rigorous to me. However, the similarities between the new implementation and existing methods should be clarified. Additionally, the real-world trade-offs remain unclear. The experiments in Figures 6 and 7 do not cover all the different methods and do not consider the performance with normalization in terms of runtime. I understand that such trade-offs might not be consistent across different tasks, but I believe that providing more experimental results on existing real-world benchmarks comparing different methods (CGTP, GTP, MTP) with normalization would significantly enhance the contribution. I would consider raising my score if the authors address these concerns.

**Relation To Broader Scientific Literature:**

Potentially, building expressive yet efficient equivariant neural networks using tensor product operations is a challenging and important problem. I believe this paper contributes to the development of such networks and, consequently, to their application.

**Theoretical Claims:**

The theoretical claims appear to be correct overall, as well as their proofs.

---

> ### Author Rebuttal · Authors · 2025-04-01
>
> We thank the reviewer for your careful reading of our work and helpful feedback. We appreciate that the reviewer finds our measure of expressivity reasonable and our runtime evaluation rigorous and comprehensive.
>
> Regarding the weaknesses
> 1. > Contribution of the new GTP implementation appears limited
>
>      We very much agree that spherical harmonic transforms (which we refer to as $S^2$ fourier transforms) have been extensively studied and we cited Healy et. al. 2003 which improves upon the seminal work by Driscoll and Healy 1994 (which we will add as a citation). However, much of the work on S2FFTs involves numerical precision issues for super high $\ell$s when using asymptotically fast algorithms. This is beyond the scope of our work as most equivariant networks are still limited to small $\ell$. Instead, we just point out asymptotic improvements exist if equivariant networks scale to super high $\ell$ in the future. Importantly, we emphasize that this connection gives the first asymptotic runtime improvement over sparse CGTP that is not simply a consequence of reduced expressivity.
>
>      Our actual grid implementation uses a much simpler seminaive algorithm described in Healy et. al. 2003 and Dricoll and Healy 1994.
>
> 2. Our experiments in Section 6.1 (Figure 6) mainly serve to show a drop-in replacement of Fourier GTP with Grid GTP leads to speedups. These are done on standard benchmarks of 3BPA and rMD17 and with an equivalent number of GPU hours. If the reviewer is concerned about actual training performance, comparisons against CGTP can be found in the original GTP paper (https://arxiv.org/abs/2401.10216).
>
>      Our experiment in Section 6.2 (Figure 7) is a simple demonstration that lack of “antisymmetric” interactions can have tangible limitations on model performance. If the reviewer is concerned about how the lack of such interactions affect real world datasets, which is an excellent question, we actually strongly suspect most common molecular datasets are minimally impacted. Loosely speaking, this is because irrep types of commonly predicted quantities such as energy or forces can be constructed purely from symmetric tensor products of the input irrep types. We are actively exploring the impact of antisymmetric tensor products in future work.
>
> # Other Comments or Suggestions
> Your understanding is correct, the $c_*$ are meant to separate the multiplicity of irreps. For CGTP, we wrote $c_Z$ as a tuple to try to highlight which inputs contributed, however in practice, this tuple is simply embedded as an integer. We will explain this more clearly in our revised manuscript.

---

> > ### Comment · Reviewer_t24S · 2025-04-05
> >
> > Thank you for the response. I will maintain my score.

---

> > > ### Author Response · Authors · 2025-04-08
> > >
> > > We would like to thank you again for your time spent reviewing our work. Your feedback was very useful in helping improve our work. In addition, we will add our code as supplementary material in the camera-ready version if accepted.

---

### Official Review · Reviewer_a8uP · 2025-03-14

**Overall Recommendation:** 3

**Summary:**

This paper investigates tensor product operations in E(3)-equivariant neural networks, an
important class of models for 3D modeling tasks, which have been recently proposed as a
faster alternative to the standard Clebsch-Gordan tensor product. In particular, the authors
introduce measures of expressivity and interactability, and analyze the runtimes (FLOPs and
wall-clock time), GPU utilization, expressivity, and asymptotics of various operations. Finally,
they  provide  a  novel  implementation  of  the  Gaunt  tensor  product  without  sacrificing
asymptotics, which is shown to be faster in practice.

**Claims And Evidence:**

The main claim in this paper is that, although several operations have been proposed to achieve E(3) equivariance efficiently, efficiency, in many cases, comes at the cost of expressivity. This claim is supported by appropriate expressivity measures, while runtime is examined by carefully designed benchmarks illustrating discrepancies from the theoretical bounds. Additionally, the improvement they proposed is shown experimentally to be indeed a faster version of the GTP tensor operation.

**Essential References Not Discussed:**

Nothing to note.

**Experimental Designs Or Analyses:**

I found the experimental designs/analyses carefully designed, with several factors tested (asymptotic, FLOPs, GPU utilisation, wall-clock time).

**Methods And Evaluation Criteria:**

The evaluation criteria (expressivity and runtime) are reasonable, since it is a typical trade-off in equivariant machine learning. Additionally, the proposed implementation (although not strictly a new method) makes sense as it shows runtime improvements.

**Other Comments Or Suggestions:**

1.  In  lines  59-60,  the  authors  state  that  linear maps between equivalent irreps are
multiples of the identity. To my knowledge, this is not the case for real representations
of  an  arbitrary  group.  However,  this  does  hold  by  Schur’s  lemma  for  complex
representations.

2.  Typo in lines 249-250: “(some details about the hardware here)”.

3.  Typo in Equation (19).

**Other Strengths And Weaknesses:**

**Strengths**

1.  The paper is well-organized and well-presented (although I believe that some prior knowledge of the field is required to carefully follow it).
2. The experimental section is thorough, clearly delivers evidence for the claims made, and illustrates the differences across different tenor product operations.
3.  The paper provides important clarifications and clears possible misunderstandings regarding the operations that have been proposed in the literature (Table 1 is quite useful). Additionally, it provides important insights to practitioners by thoroughly analyzing the asymptotics and expressivity of various tensor product implementations. These insights can help pave the way for developing and analysing new tensor product operations.

**Weaknesses**

1.  Normalizing runtimes for expressivity does not seem sufficiently justified, or more precisely, does not necessarily give sufficient guidance to practitioners on what to choose. For example, generalisation or optimisation might be equally or more crucial for certain real-world tasks, and therefore it might be the case that the cheaper operation should be chosen.
2. Although the paper makes important clarifications for the field, I am a bit concerned that it lacks novelty, since it mainly analyses existing methods without necessarily providing actionable guidelines.
3.  It is not evident how much of a fair comparison is made in benchmarking the various tensor product implementations. Could it be the case that different implementations might (significantly) alter the experimental metrics (as with GTP)?

**Questions For Authors:**

1.  Can you justify the contribution of normalizing for expressivity? While the defined
measure  of  expressivity  and  asymptotic  runtimes  make  sense  on  their  own,  for
example, the conclusion about the ratio runtime/expressivity is not really clear.

2.  In  Figure  4,  what  is  the  reason  for  CGTP  (Sparse) having low FLOPs but high
walltime?

3.  Have  you  considered  any  examples  of  real-world  datasets  where  leveraging
antisymmetric  interactions would improve performance? It would be interesting to
evaluate the various tensor product operations on such a problem.

**Note**. The paper is technically sound and provides certain valuable insights. I am currently (hesitantly) leaning towards acceptance, but I hold reservations due to my concerns about novelty, and perhaps also relevance/impact (since the improvements proposed are mostly related to implementation).

**Relation To Broader Scientific Literature:**

E(3)-equivariant neural networks are an important class of models for 3D modeling tasks,
with the tensor product being the key non-linear operation, in several such architectures.
These have found application in, e.g. molecular modeling and physical simulations.
The authors consider recently proposed several alternatives to the standard Clebsch-Gordan tensor
product  in  the  literature,  which  offer  improved  runtimes, thus well-contextualising their work.

**Theoretical Claims:**

I have looked at the proof of Theorem 3.2, which appears correct.

---

> ### Author Rebuttal · Authors · 2025-04-01
>
> We thank the reviewer for the thorough feedback on our work. We appreciate that the reviewer finds our work well-organized and presented, the experiments thorough, and provides important insights to practitioners.
>
> Regarding the weaknesses
> ## Weaknesses
> 1. This is a great question. The motivation is to highlight the difference in runtime improvements caused by cleverly reducing degrees of freedom from improvements caused by actually making tensor products faster. In particular, we also emphasize that the output spaces of these algorithms are different, making a direct comparison unfair.
>
> 2. We provide a framework to analyze the expressivity-vs-runtime tradeoffs in popular tensor product operations (TPOs) in equivariant neural networks. These tradeoffs are non-obvious, and can help other practitioners realize that these operations are not necessarily equivalent to each other. Importantly, we highlight that our work has revealed that most existing TPOs actually only get improvements from reducing degrees of freedom. Our connection to $S^2$ fast Fourier transforms provides the first (though for now impractical) algorithm which has an asymptotic improvement on the runtime/expressivity ratio.
>
>      Our microbenchmarking efforts help highlight the different algorithmic tradeoffs. MTP focuses on maximising GPU utilization with a more matrix-multiplication friendly algorithm which ends up costing more FLOPs and hence higher overall runtime. CGTP on the other hand is FLOPs efficient but suffers from poor GPU utilization. GTP offers a balance between both.
>
>
> 3. This is a great question as indeed, different implementations could have drastically different walltimes in practice and we tried to mitigate implementation differences as much as possible (e.g. implementing all of the algorithms in JAX). We would like to highlight that it is not the results themselves but the method of benchmarking that is important.
>
>      Our microbenchmarks provide valuable insight on how different implementations can be improved. In particular, the discrepancy between FLOP counts (which are invariant) and walltime indicates potential for significant acceleration (eg. custom kernels) by better utilizing the GPU.
>
> ## Other comments and suggestions
> 1. Good catch! Indeed we do need an algebraically closed field for the maps to be identity. In the case of $SO(3)$, the irreps over complex vector spaces are real irreps.
> 2. Thanks for catching this! We meant to add 'AMD EPYC'.
> 3. Thanks for catching this!
>
> ## Questions
> 1. As discussed above, the normalization for expressivity accounts for the fact that the output spaces of the different TPOs are not identical. We agree that there can be many ways to account for this difference.
> 2. Good question! The algorithm we used for CGTP-sparse (Appendix G) while being able to exploit the sparsity, introduced a lot of overhead due to the conditional logic leading to poor GPU utilization. As is seen in recent efforts (https://developer.nvidia.com/blog/accelerate-drug-and-material-discovery-with-new-math-library-nvidia-cuequivariance/, https://arxiv.org/abs/2501.13986v2), a more GPU friendly algorithm can improve the runtime.
> 3. This is an excellent question! In most of the popular datasets, we strongly suspect antisymmetric operations to have minimal impact. Loosely speaking, this is because irrep types of commonly predicted quantities such as energy or forces can be constructed purely from symmetric tensor products of the input irrep types. We plan to explore the impact of antisymmetric tensor products further in future work.

---

### Decision · Program_Chairs · 2025-05-01

**Decision:**

Accept (poster)

**Comment:**

This paper had 3/4 reviewers recommending acceptance. Reviewers appreciated the well-supported claims, quality of presentation, and insights about several tensor product operations used in equivariant NNs.

The paper has one contribution towards improving E(3)-equivariant models, which is a simpler and faster  implementation of the Gaunt tensor product (GTP). This was considered a small contribution by the reviewers.

The main contribution claimed is the method for analyzing and comparing different tensor product operations in the equivariant NN literature. Most reviewers found this contribution valuable and that it provides useful insights about prior work. Reviewer Hd64 questioned whether the methodology is useful in practice and raised several potential issues with the implementation details and metrics. In the back-and-forth with the authors, the reviewer raised their score but not to the point of acceptance.

I side with the majority here. Even if the method to compare and evaluate prior work is not ideal as raised by Hd64, it seems to be the first of its kind and brings useful insights to researchers and practitioners on the field. I believe it is worth being published.